# A single nuclear transcriptomic characterisation of mechanisms responsible for impaired angiogenesis and blood-brain barrier function in Alzheimer's disease

Stergios Tsartsalis [1,2,12], Hannah Sleven[3,12], Nurun Fancy [1,4,12], Frank Wessely[5], Amy M. Smith[1,4,6], Nanet Willumsen[1,4], To Ka Dorcas Cheung[1,4], Michal J. Rokicki[5], Vicky Chau[4], Eseoghene Ifie[7], Combiz Khozoie[1,4], Olaf Ansorge [7], Xin Yang [1,8], Marion H. Jenkyns [1], Karen Davey[1,4], Aisling McGarry[1,4], Robert C. J. Muirhead[1,4], Stephanie Debette [9], Johanna S. Jackson [1,4], Axel Montagne [10], David R. Owen [1], J. Scott Miners[11], Seth Love[11], Caleb Webber [5], M. Zameel Cader[3] & Paul M. Matthews [1,4,8] ✉

Brain perfusion and blood-brain barrier (BBB) integrity are reduced early in Alzheimer's disease (AD). We performed single nucleus RNA sequencing of vascular cells isolated from AD and non-diseased control brains to characterise pathological transcriptional signatures responsible for this. We show that endothelial cells (EC) are enriched for expression of genes associated with susceptibility to AD. Increased β-amyloid is associated with BBB impairment and a dysfunctional angiogenic response related to a failure of increased pro-angiogenic HIF1A to increased VEGFA signalling to EC. This is associated with vascular inflammatory activation, EC senescence and apoptosis. Our genomic dissection of vascular cell risk gene enrichment provides evidence for a role of EC pathology in AD and suggests that reducing vascular inflammatory activation and restoring effective angiogenesis could reduce vascular dysfunction contributing to the genesis or progression of early AD.

Alzheimer's disease (AD), the most common form of dementia[1], is characterised by extracellular deposits of toxic forms of β-amyloid (Aβ), intracellular neurofibrillary tangles (NFTs) and neurodegeneration. Large-scale genomic association studies have suggested specific molecular processes responsible for susceptibility to disease[2–4]. The

non-neuronal cells in which these genes are expressed together are candidates for "causal" roles in the initiation of AD pathology[5].

The brain microvasculature appears to play a major role in AD pathophysiology[6–8]. Endothelial cells (EC) contribute to the clearance of Aβ and other toxic species from the central nervous system (CNS),

[1]Department of Brain Sciences, Imperial College London, London, UK. [2]Department of Psychiatry, University of Geneva, Geneva, Switzerland. [3]Nuffield Department of Clinical Neurosciences, Kavli Institute for Nanoscience Discovery, Dorothy Crowfoot Hodgkin Building, Sherrington Road, University of Oxford, Oxford, UK. [4]UK Dementia Research Institute Centre, Imperial College London, London, UK. [5]UK Dementia Research Institute Centre, Cardiff University, Cardiff, UK. [6]Centre for Brain Research and Department of Pharmacology and Clinical Pharmacology, University of Auckland, Auckland, New Zealand. [7]Neuropathology Unit, Nuffield Department of Clinical Neurosciences, University of Oxford, Oxford, UK. [8]St Edmund Hall, University of Oxford, Oxford, UK. [9]University of Bordeaux, Inserm, Bordeaux Population Health Research Center, Team ELEANOR, UMR 1219, 33000 Bordeaux, France. [10]Centre for Clinical Brain Sciences, and UK Dementia Research Institute, University of Edinburgh, Edinburgh EH16 4SB, UK. [11]Dementia Research Group, University of Bristol, Bristol, UK. [12]These authors contributed equally: Stergios Tsartsalis, Hannah Sleven, Nurun Fancy. ✉e-mail: p.matthews@imperial.ac.uk

regulate the selective exclusion of potentially inflammatory or toxic blood proteins from the brain and facilitate immune cell trafficking from blood[9]. EC and pericytes (PC) together contribute to the regulation of brain perfusion, endothelial permeability and immune activation[7,10]. Multiple in vivo imaging and *post mortem* neuropathological studies, as well as studies of preclinical models, provide evidence for chronic tissue hypoxia, impaired regulation of cerebral blood flow and maintenance of the integrity of the blood-brain barrier (BBB) in early AD and with increased CNS expression of Aβ[11–16]. Recent work has begun to elucidate transcriptional mechanisms for this[17–20].

In this work, we perform an integrated analysis of our own RNA sequencing of over >100,000 brain vascular nuclei with prior data[21] to define the enrichment of brain microvascular cells for the expression of AD risk genes quantitatively as a test of the potential causal contribution of these cells to disease genesis[5]. We explore functional roles of AD risk genes in the brain microvasculature by assessing functional enrichment of genes co-expressed with them. Differential expression and gene co-expression analyses are used to characterise genes and pathways affected with AD. We validate pathways that we discovered by re-analysis of other, previously published datasets. Finally, we explore interactions between vascular cells that could be responsible for impaired vascular homoeostasis in AD using cell-cell communication analysis. Together, our results provide a transcriptomic mechanistic description of how increasing brain Aβ with AD may impair regulation of perfusion, reduce its own vascular clearance, impair adaptive angiogenic responses and lead to loss of BBB integrity.

## Results

### Endothelial cells are enriched in genes associated with genetic risk for AD

Our analyses were based on data from 77 cortical brain samples from donors with AD (*n* = 41) or non-diseased controls (NDC, *n* = 36). Three different datasets were analysed jointly: two of the datasets included nuclei from samples in which fluorescence-activated sorting (FACS) was performed to remove neuronal and oligodendrocyte nuclei before barcoding and sequencing. This achieved a better representation of the less abundant vascular cell types of interest[21,22]. The third dataset included EC obtained by a dextran gradient-based enrichment of lightly dissociated cells to select for microvessel-associated nuclei.

After integration of the FACS-enriched datasets using LIGER[23], cells were clustered and represented in a UMAP[24] (Fig. 1A). AD and NDC donor nuclei and nuclei from different datasets, brain regions and sexes were well-mixed after integration (Figure S1). Numbers of nuclei recovered did not differ significantly between the AD and NDC samples. Feature plots of canonical cell markers identified major non-vascular cell types in the integrated dataset (Figures S2 and S3).

Cell types were identified using cluster marker identification and inspection of the expression of known cell type marker genes. EC were identified by marker genes *PECAM1*, *FLT1*, *VWF*, *NOSTRIN*, *CLDN5* and *IFI27*[19,25] (Figs. 1B and S4A). Specific expression of *COL1A1*, *COL12A1*, *COL6A1* and *COL5A1* was used to identify fibroblasts (FB) (Fig. 1B and S4B). To distinguish PC from SMC nuclei, we re-clustered the EC, FB and vascular mural cell (PC and SMC) nuclei from the total dataset, as shown in the UMAP plot of Fig. 1C; separate, heterogeneous clusters of vascular mural cell nuclei expressed *PDGFRB* and *RGS5 (*characteristic of PC[19]) and *ACTA2* (highly expressed in smooth muscle cells (SMC)[19]) (Fig. 1C and S4C). This re-clustering allowed us to separate the SMC, confirmed by the high expression of *ACTA2* and *MYH11* with very low levels of *RGS5* and *GRM8* from the PC (Fig. S5). The total dataset included 70537 EC, 20885 FB, 9594 PC and 971 SMC nuclei.

We confirmed our cluster annotations (Supplementary File 1) by demonstrating significant mutual overrepresentations of our cluster markers and those reported previously in human[17,19,26] (Fig. 1D–F) and mouse[25] (Fig. 1G) single nuclei or single cell RNA sequencing studies. We found EC were most enriched for capillary

markers defined by previous studies, but those associated with arterial and venous zones also were represented (Fig. S6A–C). Based on previously described meningeal and perivascular FB markers[17], we showed that our FB nuclei were most enriched for perivascular FB markers: 77/80 of our top FB markers overlapped with those for perivascular FB (Fisher's exact test (FET) for overrepresentation, $p = 2.82 \times 10^{-77}$); only 38/80 of our top FB markers overlapped with those for meningeal FB ($p = 6.36 \times 10^{-19}$).

To exclude any significant contamination of the vascular cell clusters by microglia or astrocytes, we assessed their enrichment in gene expression sets that specifically characterise microglia and astrocytes using Expression-Weighted Cell type Enrichment (EWCE). The microglia[27]- and astrocyte[28]-specific gene sets were uniquely enriched in the corresponding cell clusters; clusters corresponding to the vascular nuclei showed no significant enrichment (Fig. S7).

Well-annotated genes associated with genetic risk of AD[2–4] (Supplementary File 2) were expressed in nuclei from all four vascular cell types (Fig. 1H): 52/61 AD risk genes tested were found in at least one of the vascular cells studied, although less than half of these genes were expressed in 5% or more of nuclei (EC, 21/61; FB, 21/61; SMC, 17/61; PC, 19/61). 14/61 of these genes were expressed in at least 5% of the nuclei across all *four* cell types (*ADAM10, APOE, CD2AP, CELF1, CLU, CNTNAP2, FERMT2, IQCK, MEF2C, PICALM, SORL1, SPPL2A, USP6NL, WWOX*).

We employed MAGMA.Celltyping to test for the significance of the enrichment of vascular nuclei across the larger set of genomic loci associated with AD[5]. First, we generated a dataset of nuclei that were directly processed for snRNAseq, thus it included all the canonical cell types of the brain (total brain nuclei dataset) (Figs. S8–S10). This confirmed that the AD risk gene expression enrichment in the brain was greatest in microglia, as reported previously[5] (Fig. 1I, dark brown). Vascular cells showed enrichment similar to that seen with oligodendroglia and astrocytes. We then repeated the analysis after controlling for the microglial enrichment. The independent gene set enrichment of vascular cells and oligodendroglia decreased substantially for the vascular cells and oligodendroglia, suggesting expression of similar AD risk gene sets (Fig. 1I, dark green). To partition enrichment amongst the individual vascular cell types, the analysis was repeated with vascular cell data alone. Only EC were significantly enriched for expression of AD risk genes (Fig. 1J, dark brown). Together, these results suggest that the molecular mechanisms associated with the genetic risk for AD are expressed in EC, but largely overlap with those that also confer risk expressed in microglia.

Vascular cells are uniquely enriched for expression of genes associated with white matter hyperintensities (WMH) (Fig. S11A). A similar analysis across the individual vascular cells (EC, FB, SMC, PC) suggested similar enrichments across EC, FB and SMC (Fig. S11B). We then questioned whether the brain vascular AD genetic risk enrichment of EC could be explained by genes shared with small vessel disease, a common co-morbid pathology[29]. To answer this, we re-estimated AD risk gene enrichment in vascular cells after statistically controlling for that associated with WMH[29]. The results remained virtually unchanged (Fig. 1J, light brown). These data therefore imply that EC in brain microvasculature play a role in AD genesis (susceptibility)[5] independent of that for any co-morbid primary cerebral small vessel disease.

### Transcriptional signatures of dysfunctional angiogenesis in AD
We tested for transcriptomic evidence for specific mechanisms of microvascular pathology in AD using a mixed-effects model in MAST[30] to discover genes differentially expressed in AD relative to NDC for each of the vascular cell types. EC showed similar numbers of up- (76) and down-regulated (86) genes with AD (Fig. 2A, B). Relatively more genes downregulated in FB (Fig. 3A, B) and PC (Fig. 3C, D) (47 genes, FB; 47 genes, PC; FDR 0.1) than upregulated (25 genes, FB; 20 genes, PC) (Supplementary File 3). We did not find significantly differentially

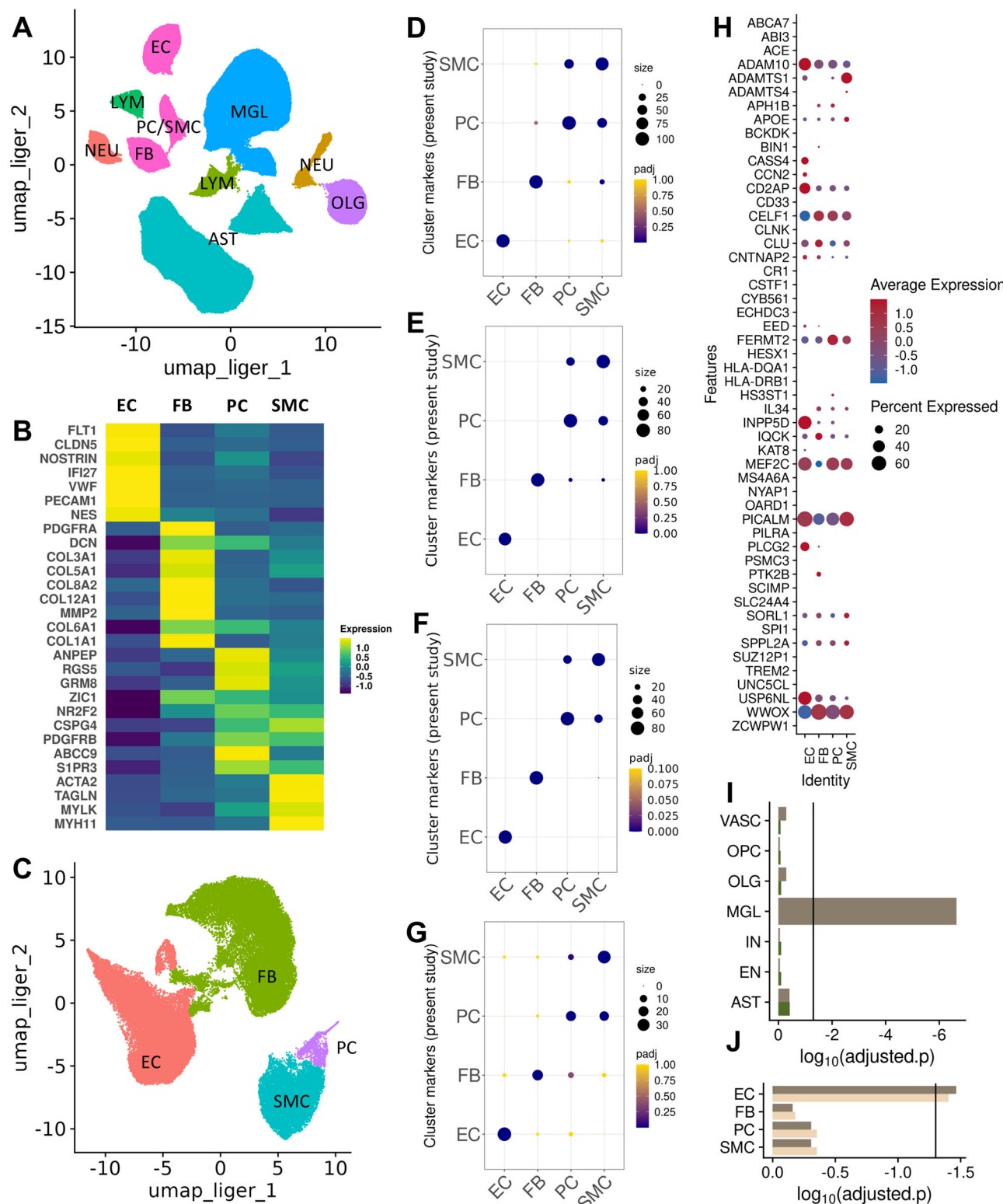

expressed genes (DEG) in the SMC, for which there were a relatively low number of nuclei available for analysis.

**Transcriptional signatures of pathological angiogenesis and blood-brain barrier integrity with AD.** Pathways involved in the angiopoetin-Tie2 signalling system involved in later stages of vascular development were differentially expressed in EC with AD. Pathways involved in regulation of endothelial cell migration and angiogenesis were upregulated (Fig. 2C) in association with increased expression of

proangiogenic *ANGPT2, HIF1A*, and *FGF2* transcripts (Fig. 2B, Supplementary File 3). Immunohistochemistry (IHC) also showed increased expression of EC ANGPT2 (Fig. 2D) and FGF2 (as well as its receptor FGFR1, Fig. 2D) with AD. However, the expression of multiple functionally related genes that are downstream effectors of angiogenic and trophic signalling in EC, such as VEGFA-VEGFR2, EGF-EGFR, TGFβ, members of the insulin and IGF1R pathways (e.g., *SPRED2, RAC1, RASAL2, TEK, DUSP16, SPTBN1, TRIO, ARHGEF7*) and genes in the NOTCH signalling pathway (*TLE4, NCOR2, ADAM10*) were

**Fig. 1 | Characterisation of cell-type specific transcriptomes and their relative enrichment in Alzheimer's disease risk genes. A** UMAP plot of the integrated snRNAseq dataset from 57 brain samples. **B** UMAP plot after re-integration and clustering of the EC, FB, PC and SMC nuclei in (**A**) for discrimination between PC (cyan) and SMC (purple) nuclei (EC, coral and FB, green). **C** Heatmap of the average scaled expression of representative marker genes for each cluster. **D–F** Dot plots of the overlap between cell markers for EC, FB, PC and SMC previously identified in human[17,19,26] and (**G**) mouse[25] snRNAseq studies and the cluster markers used in the present study. The size of the dots correspond to the overlap between the cluster gene sets and the colour of the dot to the adjusted *p* value of a one-sided over-representation Fisher's exact test. **H** Dot plot of the average scaled per cluster expression of genes previously associated with genetic risk for AD (size, percentage of nuclei per cluster with >1 count; colour scale, average scaled gene expression). **I** MAGMA.Celltyping enrichment of brain nuclei in genomic loci associated with genetic risk for AD. The bars correspond to the $\log_{10}$p-value (one-sided) of the

enrichment in GWAS signal i.e., the linear regression between cell type specificity of gene expression and the common variant genetic association with the disease using information from all genes (dark brown, line indicates significance threshold adjusted for all cell types). Enrichment of vascular nuclei is reduced after controlling for genes enriched in microglia (dark green). This analysis was performed on 153′128 nuclei from 36 independent samples (**J**) MAGMA.Celltyping AD risk gene enrichment of nuclei of the brain vasculature (dark brown bar, line indicates significance threshold adjusted for vascular cell types). Enrichment is not changed substantially after controlling for the enrichment of genetic loci associated with white matter hyperintensities (WMH) (light brown). This analysis was performed on 51′874 nuclei from 57 independent samples Abbreviations: AST astrocytes, EC endothelial cells, FB fibroblasts, MGL microglia, NEU neurons, NEU neurons, OLG oligodendrocytes, PC/SMC pericytes and smooth muscle cells, LYM lymphocytes. Source data are provided as a Source data file.

downregulated. Upregulation of *ANGPT2*, although it may stimulate angiogenesis, also can increase vascular leakage and instability by inhibiting the Ang1-TIE2 pathway[31–33]. Perivascular FB nuclear transcriptomes showed downregulation of VEGF, FGF, EGF and IGF pathway genes (including *SPRED2*, *DAB2IP* and *SPTBN1*), *DTX2*, a regulator of Notch signalling[34] and *LAMC1*, which encodes for a component of the ECM (Supplementary File 3). Angiogenic pathways including those for TGF-β, NOTCH and Wnt/β-catenin signalling supporting blood-brain barrier integrity[35] and adaptive immune activation pathways (including genes associated with "positive regulation of memory T cell differentiation", "positive regulation of T cell chemotaxis") were upregulated with AD in PC (Supplementary File 3).

To identify additional functional pathways associated with individual genes differentially expressed in vascular cells with AD, we performed gene co-expression analysis for EC, FB and PC in pooled AD and NDC samples (MEGENA[36], Supplementary File 4). We reasoned that pathways functionally altered in AD would show a statistically significant overrepresentation of DEG in AD (Supplementary File 5). The co-expression network structure for EC (Fig. 2E), FB (Fig. 3E) and PC (Fig. 3F) describes relationships between modules significantly enriched in genes differentially expressed with AD (modules enriched for genes upregulated with AD, red; modules enriched for genes downregulated with AD, blue). Downregulation of co-expression network modules enriched for TGF-beta, EGFR, angiogenesis, Notch signalling, focal adhesion, adherens and tight junction gene pathways encoding proteins involved in the maintenance of BBB with AD (e.g., *RAC1*, *RASAL2*[37]) in association with increased expression of Module 1287 enriched for interleukin-/NFκB and interleukin-2 pathways suggests an inflammatory trigger for loss of BBB integrity (Fig. 2F). Adherens junction and cadherin binding genes also were enriched in FB (module 2, Fig. 3G) and PC (module 19, Fig. 3H) modules downregulated with AD. PDGF and EGFR signalling was enriched in the downregulated PC module 19. Gene co-expression modules in FB functionally related to vascular homoeostasis were significantly enriched in the DEG downregulated with AD, e.g., Module 2, which is significantly enriched in NOTCH signalling genes (e.g., *NOTCH1* and *NOTCH2* and regulators of NOTCH expression[34], *ARRB1* and *DTX2*; Fig. 3G, Supplementary Files 4 and 5). Module 2 was significantly enriched in genes (*SPRED2*, *DAB2IP*, *ARRB1*, *SPTBN1*) encoding for downstream genes in vascular growth factor signalling pathways (e.g., FGFR1-4, VEGFR2, EGFR) and genes for ECM proteins (*COL5A1* and *COL1A2*)[38]. Similarly, FB Modules 3, 132 and 408 were enriched for upregulated DEG including *ANGPT* and *TGFB2*, the protein products of which are involved in angiogenic signalling to EC, as well as solute carrier genes involved in nutrient and metabolite transfer across the blood-brain barrier[39]. However, like EC, VEGFA-VEGFR2 and TGFβ pathways were represented in downregulated co-expression modules. We also found downregulation of EGF/EGFR signalling pathways in PC (genes for which include *RPS6KA2*, *ASAP1*, *MEF2D* and *EGFR*, Fig. 3H,

Supplementary File 3), an expected consequence of downregulation of VEGF-VEGFR2 signalling[40].

To validate these findings, we tested for their reproducibility in other published datasets (Supplementary File 6) by assessing whether gene sets representing the functional pathways that were enriched in our DEG overlapped with genes differentially expressed with AD in these independent datasets using gene set enrichment analysis (fgsea package v1.22.0)[41]. The fgsea package performs a gene set enrichment analysis in which quantitative parameters (in our case, the product $\log_{10}$(pval)*logFC of the DGE analysis on the independent datasets) is used to weigh the genes on which the enrichment is calculated. First, we tested differentially expressed genes in EC data from a brain vascular-enriched snRNA sequencing dataset[17]. Then, we performed a differential gene expression (DGE) analysis in a bulk RNAseq dataset encompassing over 600 AD and NDC individual brain samples from three large datasets[42–44]. VEGFA-VEGFR2 and RAS pathways and adherens and tight junction genes were downregulated in the snRNAseq EC dataset[17]. Functional pathways associated with angiogenesis initiation were upregulated in both datasets. These analyses implicate impaired angiogenic signalling, dysfunctional angiogenesis and reduced trophic support, tight junction gene expression and vascular integrity in AD.

**An association between innate immune activation in EC and increased FB and PC apoptotic gene expression with AD.** DEG and co-expression module enrichment provided evidence for innate immune activation in association with increased pericyte and fibroblast apoptosis in AD. Upregulated DEG in EC were enriched for Toll-like receptor genes (*ATF1*, *MEF2C* and *RPS6KA5*) suggesting pathological activation expected to promote the recruitment and functional modulation of immune cells[45]. Interleukin (IL) and interferon (IFN) signalling genes (*SPRED2*, *IFNGR1*, *MX2*, *RASAL2*, *XAF1*, *DUSP16*, *SPTBN1*) were amongst downregulated DEG (Supplementary File 3). Co-expression modules enriched for IL6 signalling genes (e.g., *IL6R*, *IL6ST*, *JAK1*, *JAK3* and *STAT3*) also were downregulated (Fig. 2F, Supplementary File 5).

An association between innate immune and pro-apoptotic gene expression with AD was found in FB (Fig. 3G), in which module 3, which is enriched for Toll-like receptor, IL and IFN signalling genes, was upregulated, while the anti-apoptotic *BCL2L1* was significantly downregulated (Supplementary File 5). Co-expression analyses also suggested a gene programme for increased PC apoptosis with downregulation of the negative regulator of apoptosis and inflammation, *CFLAR*, representation of the pro-apoptotic and pro-inflammatory *RIPK2* in the upregulated module 7 and identification of the anti-apoptotic *NCOA3*[46] in downregulated co-expression module 19. Module 7 in PC (Fig. 3H, Supplementary File 5), which was significantly enriched in upregulated DEG, included Toll-like response genes *MEF2A*, *PPP2CB*, *MEF2C*, *RIPK2* and *IRAK2*. These analyses

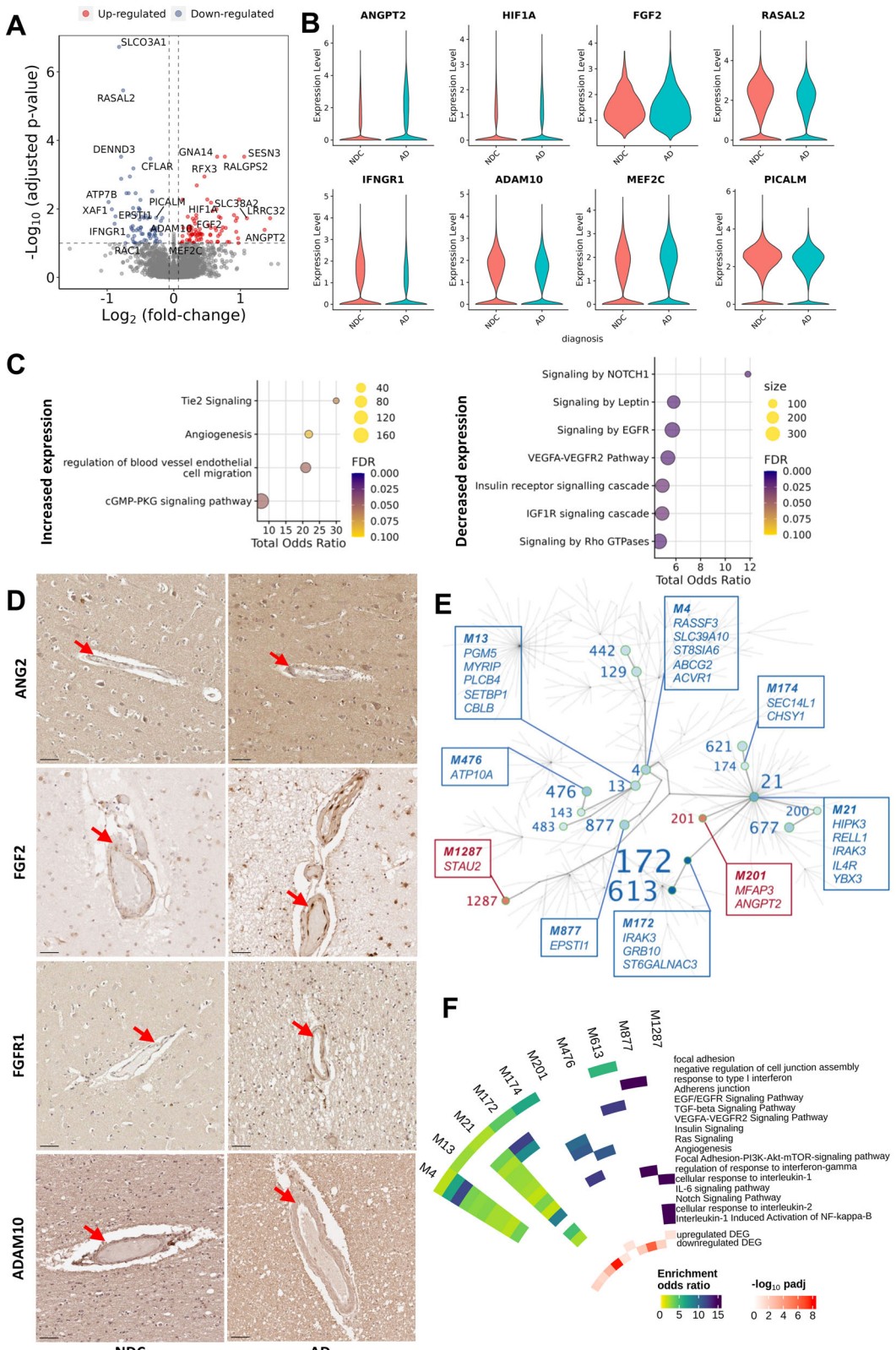

together thus provide evidence for increased apoptosis of FB and PC in AD accompanied in association with innate immune activation of EC.

### Functional relationships between dysfunctional angiogenesis and abnormal vascular amyloid processing

The directions of changes in expression of GWAS risk genes related to amyloid processing and angiogenesis in AD differed between EC and

PC. Both *PICALM*, which encodes a clathrin assembly protein modulating clearance of Aβ, and *ADAM10*, an α-secretase catalysing both non-amyloidogenic pathway Aβ precursor protein and NOTCH processing[47,48] were downregulated in EC with AD. *MEF2C*, a negative regulator of angiogenesis[49], was upregulated. By contrast, *ADAM10* and the inhibitory complement receptor *CD46* gene were upregulated in PC with decreased expression of *IRAK3* (Supplementary File 3).

**Fig. 2 | Alzheimer's disease is associated with dysregulation of vascular homoeostasis in EC. A** Volcano plot showing genes differentially expressed in AD relative to NDC donor cortical tissue in EC. Representative significantly differentially expressed genes are identified. **B** Violin plots of representative genes differentially expressed in EC with AD relative to NDC. *ANGPT2* (logFC=1.46, padj=0.04), *HIF1A* (logFC=0.68, padj=0.02), *MEF2C* (logFC=0.28, padj=0.09) and *FGF2* (logFC=0.34, padj=0.05) are significantly upregulated, whereas *RASAL2* (logFC = −0.76, padj=3.48×10⁻⁶), *IFNGR1* (logFC=0.89, padj=0.03), *ADAM10* (logFC = −0.30, padj=0.06) and *PICALM* (logFC = −0.27, padj=0.02) are downregulated. Statistical significance was determined using a likelihood ratio test with a mixed-effects model and a zero-inflated negative binomial distribution (two-sided). For demonstration purposes, the *FGF2* violin plot describes expression only for nuclei in which *FGF2* is expressed, although the statistical analysis was performed on all nuclei. **C** Dot plots of the functional enrichment analysis on the DEG that are up- and down-regulated in EC (dot size, functional enrichment gene set size; colour, FDR, one-sided over-representation Fisher's exact test) with AD relative NDC. **D** IHC of sections from the somatosensory cortex of NDC (left) and AD (right) donors highlighting increased expression of ANG2 (coded by *ANGPT2*), FGF2, FGFR1 and decreased expression of ADAM10 in the vessel wall with AD. Arrowheads denote the protein binding in the vascular wall. Scale bar = 50 μm. The IHC experiment was performed on 24 independent samples. **E** Gene co-expression module hierarchy for EC. Modules that belong to the same branch are related, i.e., larger ("parent") modules are closer to the centre of the plot and are further divided into subset ("children") modules. "Children" modules are subsets of the "parent" ones and have higher numbers as names than their "parents". Modules that show a significant overrepresentation of DEG (as shown in the volcano plots of Fig. 2A) by means of a (one-sided) Fisher's exact test are labelled and represented as coloured points in the graph (red, for modules showing an overrepresentation of upregulated DEG; blue, showing an overrepresentation of downregulated DEG). Module number font size corresponds to the significance of the overrepresentation of DEG in the module. In the boxes, the top (maximum 5) hub genes (genes with the higher number of significant correlations within the module) are described. **F** Circular heatmap of odds ratios from the functional enrichment analyses for the EC modules that show a significant DEG overrepresentation (significant modules that show redundant functional enrichment terms were omitted from this heatmap). The adjusted *p* values of the significance of the overrepresentation are provided in Supplementary file 5. The inner two tracks of the circular heatmap represent the significance (−log₁₀(padj), one-sided) of the overrepresentation of down- (innermost track) and up-regulated DEG (second innermost track). The DGE and co-expression analyses were performed on 70'537 nuclei from 77 independent samples. Source data are provided as a Source data file.

We expanded our exploration of cell-specific risk gene pathways by identifying genes in co-expression networks that were either directly connected to an AD risk gene or connected with an AD risk gene through at most one other gene (termed GWAS gene "neighbourhoods")[36,50] (Supplementary File 7). Determining the overrepresentation of genes differentially expressed in AD vs NDC in the neighbourhoods of each GWAS gene in the cell-specific co-expression networks allowed us to extend our description of pathways relevant to mechanisms of dysfunctional angiogenesis. Figure 4A highlights genes in the neighbourhoods of AD GWAS genes showing significant overrepresentation of differentially expressed in AD compared to NDC. The largest neighbourhood was associated with downregulated *PICALM* (which facilitates transcytosis and clearance of Aβ across the BBB[51,52]) in the EC co-expression network (Fig. 4A). Pathway enrichments for the largest gene neighbourhoods in EC are shown in Fig. 4B: the *PICALM* neighbourhood was enriched for NOTCH (*TNRC6C, B4GALT1, TFDP2, POFUT1, MAMLD1, TNRC6A*), semaphorin (*SEMA5A, ARHGEF11, SEMA6D, ITGA1, MYH11, PLXNC1*) and IL6 signalling gene pathways (*IL6ST, STAT3, JAK1*), suggesting a relationship between Aβ clearance, cytokine expression and angiogenesis[53]. The neighbourhood of *CCN2* (*CTGF*), encoding connective tissue growth factor, an extracellular matrix protein, was enriched in angiogenesis (*SEMA6A, RGCC*) and apoptosis-related genes (*RGCC, TJP1*) (Supplementary File 7).

Relationships between pathological angiogenic and inflammatory regulation in AD were highlighted further by gene enrichments in neighbourhood of other AD risk genes that we did not find to be differentially expressed. The EC neighbourhood of *SORL1*, encoding a regulator of endosomal trafficking, was enriched in genes for pathways involved in immune response (e.g., T-cell activation, TYROBP causal network and cytokine response and interferon signalling pathways) (Supplementary File 7). In FB, the neighbourhood of *WWOX* (Fig. 4A, C) was enriched in Notch signalling, collagen formation and Rac1 activation pathways in FB and both this neighbourhood and that of *IQCK* were enriched in ECM-related pathways. In PC, the neighbourhood of the *MEF2C* transcription factor (108 genes, one of the largest) included genes involved in Toll-like receptor (e.g., *MEF2A, PPP2CB, MEF2C* and *RIPK2*), semaphorin interactions and in EGF/EGFR signalling (*EPS8, MEF2A, MEF2C, PLCE1, RAF1*) pathways (Fig. 4A, D, Supplementary File 7). These analyses provide evidence that differential expression of risk genes (*PICALM, ADAM10, MEF2D, CD46* and *IRAK3*) with AD contribute to immune activation, angiogenic and apoptosis-related transcriptomic vascular pathologies.

## Evidence that the reduced perfusion, EC apoptosis, dysfunctional angiogenesis and consequent impairment of the BBB in AD are related to Aβ and pTau pathology

We hypothesised that expression of Aβ and pTau are related directly to AD-associated vascular pathology. To test this transcriptomically, we characterised the differential gene expression with increasing loads of Aβ or pTau pathology determined by quantitative IHC of tissue sections from the same brains and brain regions. We limited our regression analysis to the EC, the most abundant of the cell populations, to minimise Type I errors. We found 64 genes were significantly (adjusted $p < 0.1$) differentially expressed with greater brain regional Aβ and 241 genes differentially expressed with greater pTau immunostaining density (Supplementary File 8). Pro-apoptotic genes *RGCC*[54], *BTG1* and *AKR1C3* were amongst differentially expressed genes positively associated with increasing Aβ immunostaining density, whereas expression of the anti-apoptotic gene *CFLAR* showed a negative association (Fig. 5A, B). Expression of *SDCBP, GRN, PNP*, genes involved in the proinflammatory response was positively associated with the regional pTau immunostaining density (Fig. 6A, B). The functional enrichment of genes differentially expressed with increasing pTau in EC also suggested reduced EC proliferation and angiogenesis and increased apoptosis (Fig. 6C). Expression of *CCN2* (Figs. 5B and 6B) and of *SPARC* were positively associated with both increasing tissue Aβ and pTau immunostaining densities. Connective tissue growth factor (CTGF, encoded by *CCN2*) inhibits activation of the VEGF pathway by binding to its TSP1[55] and increased SPARC, an ECM glycoprotein, leads to sequestration of VEGF, reducing phosphorylation of the angiogenic VEGFR2 receptor and downstream signalling[56]. Genes implicated in lipid metabolism (*ACSL5, GPCPD1, SUMF1, PPARD*) were negatively correlated with increasing Aβ. Our analysis thus suggests that pathological expression of *CTGF* and *SPARC*, by opposing the effects of increased proangiogenic factor (*FGF2, HIF1A, ANGPT2*) expression, impair angiogenesis progressively in AD as pathological tissue Aβ and pTau increase.

The myelin associated glycoprotein to proteolipid protein 1 (MAG:PLP1) ratio, is reduced with hypoxia in chronically hypoperfused brain tissue[16]. We tested for this in 4 NDC and 6 AD brains (Table 2). MAG:PLP1 was reduced in the AD samples ($p = 0.012$), consistent with chronic hypoxia (Fig. 7A). This provides an index of the chronically reduced perfusion *ante mortem*. The insoluble Aβ concentration in brain homogenates also was increased ($p = 0.009$) (Fig. 7B) and we found that the ratio of MAG:PLP1 correlated inversely with insoluble Aβ ($r = −0.59$, $p = 0.06$, Fig. 7C), providing

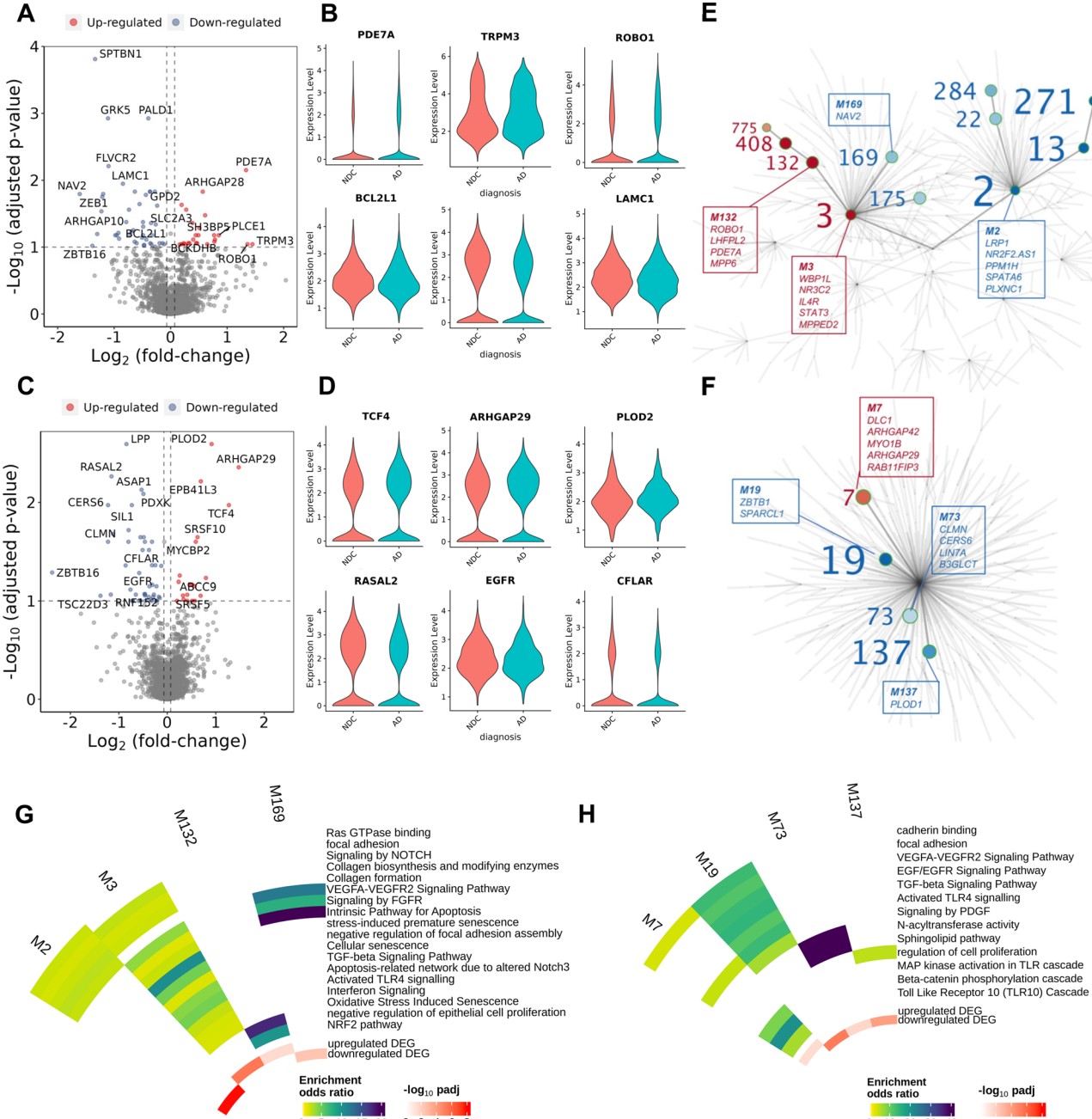

**Fig. 3 | Angiogenic and inflammatory pathways are differentially expressed in FB and PC co-expression network modules with AD.** Volcano and violin plots showing genes differentially expressed in AD relative to NDC donor cortical tissue in FB (**A**, **B**) and PC (**C**, **D**). In FB, *PDE7A* (logFC=1.33, padj=0.007), *TRPM3* (logFC=1.44, padj=0.09), *ROBO1* (logFC=1.35, padj=0.09) are significantly upregulated, whereas *BCL2L1* (logFC = −0.35, padj=0.04), *SPTBN1* (logFC = −1.34, padj=0.0001) and *LAMC1* (logFC = −0.84, padj=0.01) are downregulated. In PC, *TCF4* (logFC=1.27, padj=0.01), *ARHGAP29* (logFC=1.47, padj=0.004), *PLOD2* (logFC=0.91, padj=0.003) are significantly upregulated, whereas *RASAL2* (logFC = −1.15, padj=0.005), *EGFR* (logFC = −0.76, padj=0.07) and *CFLAR* (logFC = −0.38, padj=0.03) are downregulated. Statistical significance was determined using a likelihood ratio test with a mixed-effects model and a zero-inflated negative binomial distribution (two-sided). For demonstration purposes, the *TRPM3, EGFR, PLOD2, LAMC1* and *BCL2L1* violin plots describe expression only for nuclei in which the respective genes are expressed, although the statistical analysis was performed on all nuclei. **E** Gene co-expression module hierarchy for FB and (**F**) PC. Modules that belong to the same branch are related, i.e., larger ("parent") modules are closer to the centre of the plot and are further divided into subset ("children") modules.

"Children" modules are subsets of the "parent" ones and have higher numbers as names than their "parents". Modules that show a significant overrepresentation of DEG (as shown in the volcano plots of Fig. 3A–C) by means of a one-sided Fisher's exact test are labelled and represented as coloured points in the graph (red, for modules showing an overrepresentation of upregulated DEG; blue, showing an overrepresentation of downregulated DEG). Module number font size corresponds to the significance of the overrepresentation of DEG in the module. In the boxes, the top (maximum 5) hub genes (genes with the higher number of significant correlations within the module) are described. **G** Circular heatmap of odds ratios from the functional enrichment analyses for the FB and (**H**) PC modules that show a significant DEG overrepresentation (significant modules that show redundant functional enrichment terms were omitted from this heatmap). The adjusted *p* values of the significance of the overrepresentation are provided in Supplementary file 5. The inner two tracks of the circular heatmap represent the significance (−log₁₀(padj), one-sided) of the overrepresentation of down- (innermost track) and up-regulated DEG (second innermost track). The DGE and co-expression analyses were performed on 9'594 PC and 20'885 FB nuclei from 57 independent samples. Source data are provided as a Source data file.

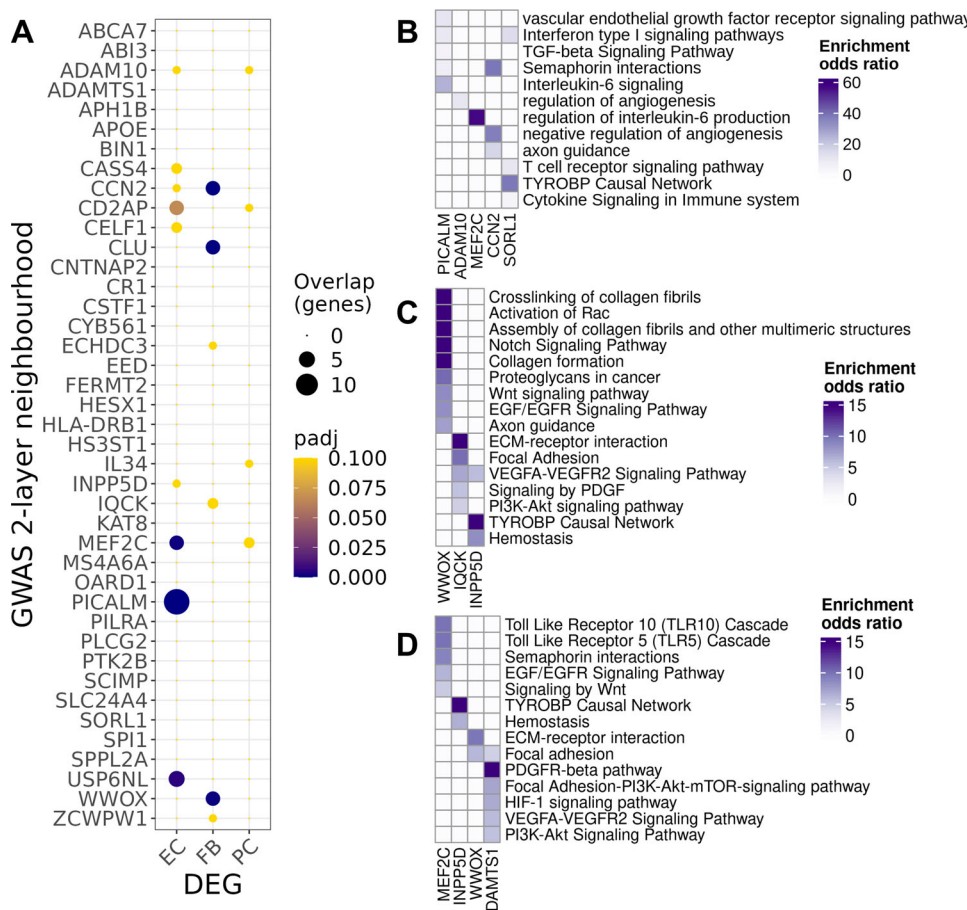

**Fig. 4 | Differentially expressed genes (DEG) with AD relative to NDC found in two-layer neighbourhoods of AD risk genes. A** Dot plot of the overrepresentation of DEG identified in each cluster (abscissa) in the 2-layer neighbourhood of each GWAS gene (ordinate) (dot size, number of the overlapping genes; colour, adjusted *p* value of the one-sided Fisher's exact test). Functional enrichment of prioritised GWAS genes in EC (**B**), FB (**C**) and PC (**D**)(colour scales represent the odds ratios of enrichment). Adjusted *p* values of the (one-sided) Fisher's exact test are provided in the source data and in Supplementary file 7. Source data are provided as a Source data file.

evidence that the chronic hypoperfusion in AD is explained in part by increased Aβ. To assess whether angiogenesis is related directly to the magnitude of local tissue hypoxia, we tested the relationship between MAG:PLP1 ratio and CD31 (an endothelial marker of vessel density) and CD105 (a marker of angiogenic endothelium)[57]. We found that the MAG:PLP1 ratio correlated inversely with CD31 (Fig. 7D) but positively with the ratio of CD105 normalised to CD31 (Fig. 7E). These data provide evidence of impaired angiogenesis in response to cerebral hypoperfusion in AD. Finally, we explored whether newly formed vessels in AD were leaky[58]. We found that local fibrinogen levels, a marker of loss of BBB integrity[59], are inversely correlated with MAG:PLP1 (Fig. 7F). Together, these data suggested dysfunctional angiogenesis and BBB leakiness in AD is related to Aβ expression and chronic cerebral hypoperfusion.

**The *APOE4* genotype potentiates EC senescence and apoptosis with AD**

*APOE4* is the AD risk gene with greatest effect size and carriers show an earlier onset of disease[60]. *APOE4* also is associated with both cerebral amyloid angiopathy and amyloid related imaging abnormalities (ARIA)[61,62]. Given the role of EC in mediating Aβ clearance, we hypothesised that EC from *APOE4* carriers (*APOE4*⁺) would show distinct transcriptomic responses with greater Aβ load. To explore this, we repeated our differential gene expression for AD in EC nuclei stratified for *APOE4* allele carriers. The analysis included only samples from donors with AD, as none of the NDC were *APOE4*⁺.

We found *APOE4*-dependent differential gene expression in EC with increasing Aβ, consistent with our hypothesis. Genes differentially upregulated in EC with increasing Aβ in *APOE4*⁺ included *HSP90AA1*, *CALM1* and *CAVIN2*[63], the protein products of which are involved in the regulation of NO synthase (NOS) (Supplementary File 8). Moreover, in *APOE4*⁻ samples, nitric oxide synthesis-related genes *CAV1* and *PTS* were downregulated with increasing pTau. A similar recent analysis of transcriptomic data from *APOE4*⁻/⁻ carrier EC with increasing Aβ described reduced expression of *PTS*, encoding 6-pyruvoyltetrahydropterin synthase, which catalyses the production of the nitric oxide synthase cofactor tetrahydrobiopterin[64]. Regression analysis of gene expression against pTau highlighted a differential expression of cytokine signalling genes with respect to the *APOE* genotype. Cytokine signalling genes were upregulated with pTau in *APOE4*⁺ nuclei but showed a negative (albeit statistically non-significant) association in *APOE4*⁻ nuclei. *APOE4*⁺ EC showed differential expression of genes involved in the positive regulation of apoptosis (*UBB, RPS7, NACC2, RPS3, SOD1*) and downregulation of the negative regulators of senescence (*ARNTL* and *TERF2*)[65,66] with increasing Aβ, whereas Aβ induced a response associated with the negative regulation of apoptosis (*PRKCA, MAP4K4*) in EC from donors who were *APOE4*⁻/⁻. These latter signatures were associated with greater expression of *CD34* and *TMSB4X*[67,68], which we interpret as evidence for increased early angiogenic pathway induction. Together, these results suggest that *APOE4* is associated with increased impairment of expression of NO signalling gene pathways necessary for regulation of

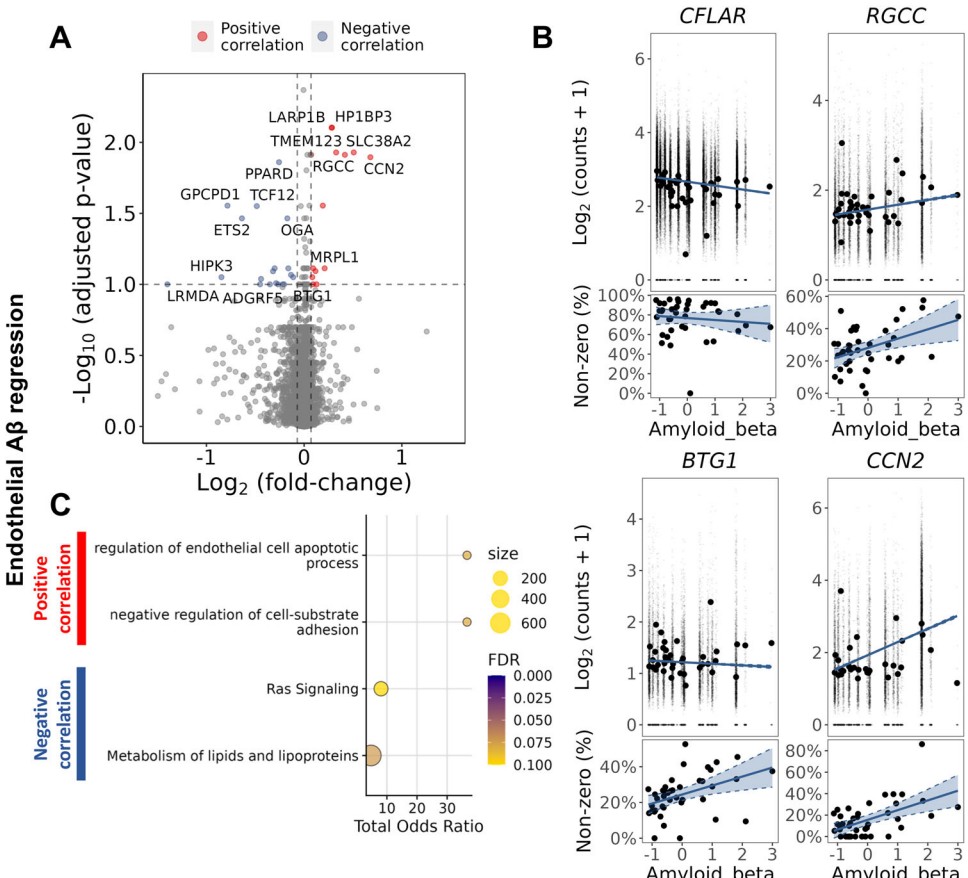

**Fig. 5 | Increased gene expression in EC with increased Aβ immunohisto-chemical staining density highlighted increased expression of genes associated with apoptosis. A** Volcano plot showing genes with a significant positive (red) of negative (blue) correlation with tissue Aβ staining density in EC.
**B** Regression plots of individual genes associated with apoptosis, illustrating their association with Aβ levels: *CFLAR* (logFC = −0.32, padj=0.08), *RGCC* (logFC=0.42, padj=0.01), *BTG1* (logFC=0.1, padj=0.09) and *AKR1C3* (logFC=0.19, padj=0.03). For each gene, two plots are presented, the plot in the upper row show the scatter plot of the regression between the average expression value and the Aβ density in each sample, whereas the plot in the lower row shows the regression between the percentage of non-zero count nuclei in each sample and the Aβ density in each sample.

The DGE analysis was performed using a likelihood ratio test with a mixed-effects model and a zero-inflated negative binomial distribution (two-sided). It takes into account both the distribution of the non-zero normalised counts (corresponding to the plot in the upper row) and the abundance of non-zero nuclei in the samples (corresponding to the plot in the lower row). Aβ in the horizontal axis is presented as scaled IHC binding values. The best-fit linear regression lines and 95% confidence intervals are shown. **C** Dot plots of the functional enrichment analysis on the DEG that are positively and negatively associated to Aβ (dot size, functional enrichment gene set size; colour, FDR, one-sided overrepresentation Fisher's exact test). Source data are provided as a Source data file.

cerebral perfusion and greater senescence and apoptosis pathway gene expression which together contribute to loss of vascular integrity with AD.

## Cell-cell interactions regulating microvascular differential expression signatures with AD

We applied NicheNet[69] to explore microvascular cell-cell interactions that could explain the molecular pathology we found in AD. NicheNet predicts interactions of ligands in sender cells with target genes (not limited to cognate receptors of these ligands but to any potential downstream gene) in receiving cells by integrating gene expression data with prior models of signalling and gene regulatory networks. Our analysis included astrocytes and perivascular macrophages (PVM) together with all of the vascular cell types as potential sender cells. To identify potential upstream regulators of the DEG in each vascular cell type, we performed NicheNet analyses, separately in each cell type, using AD DEG from EC, FB and PC to define the target gene sets (Supplementary File 9). We identified potential upstream regulators of EC DEG and assessed the cellular sources of their expression in vascular cells, PVM and astrocytes (Fig. 8A).

*GPNMB*, a gene associated with proinflammatory activation of microglia and PVM that is upregulated in AD[22], was one of the strongest predictors (and therefore a potential regulator, Fig. 8A, B) of DEG in all three vascular cell types (Fig. S12A, B). *APOE*, upregulated in astrocytes in AD (Supplementary File 3), also was among the highest ranked predictors of DEG in EC through its associations with expression of the cholesterol transport-mediated gene *ABCA1* and *RAC1*. Similarly, *VEGFA* and *TGFB1*, also were strong potential regulators of EC DEG, and were downregulated in PVM with AD. We found reduced immunostaining for VEGFA in vascular EC with AD, consistent with the downregulation VEGF signalling gene pathway expression in these cells (Fig. 8B, C). *FGF2, ANGPT1* and *ANGPT2* were other strong predictors of DEG in EC. VEGFA, FGF2, ANGPT1, ANGTP2 and TGFβ regulated many DEG in FB (Fig. S12A) and PC (Fig. S12B). *TNFSF10*, which codes for the proinflammatory TRAIL ligand expressed predominantly by EC, was implicated as regulating the majority of PC DEG, including the apoptosis regulator *CFLAR* and *EGFR*. Together, these results thus suggest that PVM and astrocyte inflammatory (GPNMB) and growth factor (VEGF, FGF and TGFβ) signalling as major regulators of AD pathology in vascular cells.

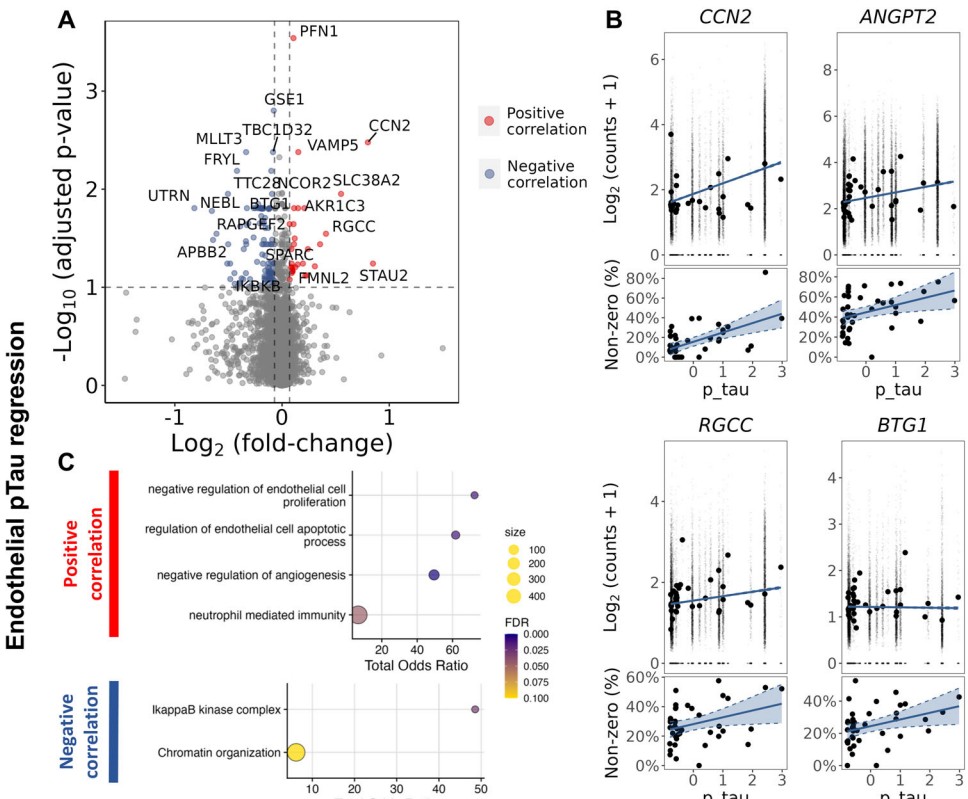

**Fig. 6 | Regression analysis of gene expression in EC as a function of pTau tissue density suggests increased apoptosis. A** Volcano plot showing genes with a significant positive (red) of negative (blue) correlation with tissue pTau density in EC. **B** Regression plots of individual genes associated to apoptosis, *CCN2* (logFC= 0.80, padj=0.003), *ANGPT2* (logFC= 1.73, padj=0.07), *RGCC* (logFC=0.41, padj=0.03), *BTG1* (logFC=0.11, padj=0.02). For each gene, two plots are presented, the plot in the upper row show the scatter plot of the regression between the average expression value and the pTau density in each sample, whereas the plot in the lower row shows the regression between the percentage of non-zero counts across the nuclei in each sample and the pTau density in each sample. The DGE analysis was performed using a likelihood ratio test with a mixed-effects model and a zero-inflated negative binomial distribution (two-sided). It takes into account both the distribution of the non-zero normalised counts (corresponding to the plot in the upper row) and the abundance of non-zero nuclei in the samples (corresponding to the lower row). pTau in the horizontal axis is presented as scaled IHC binding values. The best-fit linear regression lines and 95% confidence intervals are shown (**C**) Dot plots of the functional enrichment analysis on the DEG that are positively and negatively associated to pTau (dot size, functional enrichment gene set size; colour, FDR, one-sided overrepresentation Fisher's exact test). Source data are provided as a Source data file.

## Discussion

Previous studies have presented physiological and biochemical evidence for early and clinically significant perfusion deficits and loss of BBB integrity in AD[11–16]. Here we provide transcriptomic evidence for a central role for the brain microvasculature (and particularly EC) in the genesis of AD by demonstrating significant enrichment of AD risk genes in EC. Genes differentially expressed in microvascular cells with AD defined transcriptional signatures for dysfunctional angiogenesis with upregulated expression of genes for regulators of angiogenesis such *HIF1A, ANGPT2* and *FGF2* together with downregulation of genes for effectors of angiogenic and trophic signalling including VEGFA-VEGFR2. We also showed that the expression of multiple genes involved in maintaining the integrity of the BBB were downregulated with AD. A clue to underlying mechanisms came with transcriptomic evidence for associated innate immune activation and downregulation of potentially protective STAT3 signalling in vascular cells. Extending recent immunohistological observations related to Aβ[70], we have provided evidence that tissue Aβ and pTau are related to protein markers of chronic hypoxia and transcriptomic mechanisms for dysfunctional angiogenesis and BBB dysfunction. Finally, we showed that expression of the AD risk gene of strongest effect, *APOE4*, explains downregulation of the expression of NO synthase expression in EC, consistent with a previously proposed mechanism for reduced CBF and impaired neurovascular coupling in AD[71]. Together, these results suggest that AD risk gene expression in microvascular cells and associated inflammatory activation (explained in part by tissue levels of Aβ and pTau) are responsible for the early impairments of microvascular structure and function with AD.

A recent paper reported that GWAS risk genes were enriched in EC and vascular mural cells and suggested an "evolutionary shift" of AD risk gene expression from a singular predominance in microglia in the mouse to expression in both microglial and vascular cells in humans[17]. Our analysis extends this by showing that, among vascular cells, risk genes associated with AD are enriched significantly in EC, suggesting involvement of EC in the genesis of AD along with microglia[5]. We provided evidence that this did not arise because of co-morbid primary cerebral small vessel disease by showing that the AD risk gene set enrichment found in EC could not be attributed to overlap of enrichment in GWAS loci associated with brain white matter hyperintensities, which are markers of small brain vessel disease[72,73]. The potential joint contribution of EC and microglia to the genesis of AD was shown by demonstration that risk genes enriched for expression in EC overlapped substantially with those in microglia.

Co-expression and two-layer neighbourhood analyses provided insights into some possible functional roles for proteins encoded by AD risk genes expressed in the vascular cells. For example, we found lower expression of *PICALM* in EC with AD, suggesting that two complementary mechanisms by which vascular clearance of Aβ is reduced

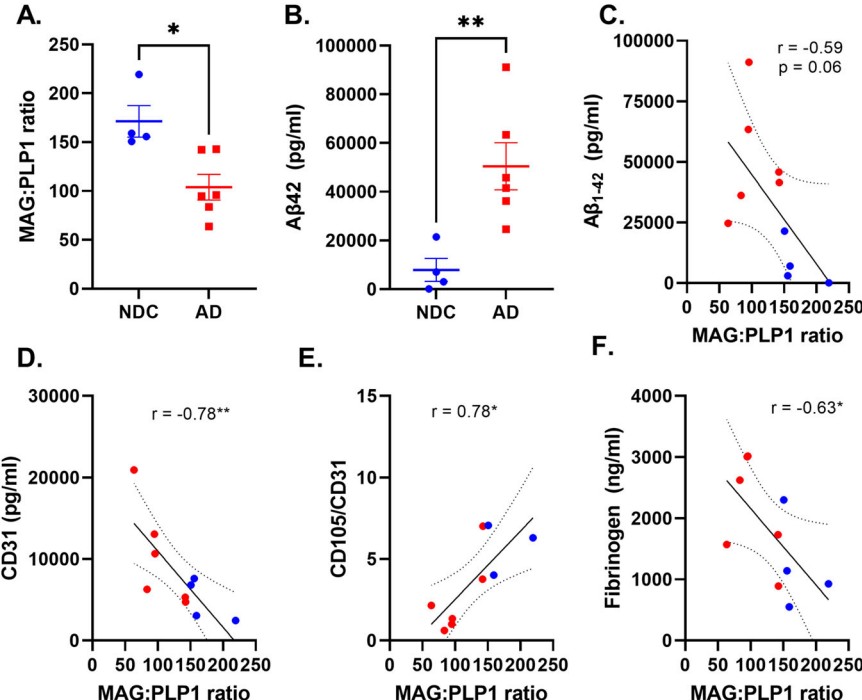

**Fig. 7 | Defective angiogenesis and blood brain barrier leakage is associated with cerebral hypoperfusion and Aβ pathology in AD. A** Scatterplot showing reduced MAG:PLP1 ratio (logFC = −0.72, unadjusted $p = 0.012$, two-sided $t$ test) in the temporal cortex with AD (red dots) relative to NDC (blue dots). Horizontal bars represent the mean ± SEM. This experiment was performed on 10 independent samples (4 NDC and 6 AD). **B** Scatterplot showing increased insoluble Aβ42 in the temporal cortex with AD relative to NDC (logFC=2.67, unadjusted $p = 0.009$). This experiment was performed on 10 independent samples (4 NDC and 6 AD). Scatterplots showing the relationship between MAG:PLP1 and Aβ (Pearson's $r = −0.59$, unadjusted $p = 0.067$, 10 samples, 4 NDC and 6 AD) (**C**), CD31 (endothelial marker) (Pearson's $r = −0.78$, unadjusted $p = 0.008$, 10 samples, 4 NDC and 6 AD) (**D**), the ratio of CD105 (a marker of neoangiogenesis) adjusted to CD31 content (Pearson's $r = 0.78$, unadjusted $p = 0.011$, 9 samples, 3 NDC and 6 AD) (**E**) and tissue fibrinogen concentration (Pearson's $r = −0.63$, unadjusted $p = 0.0049$, 10 samples, 4 NDC and 6 AD) (**F**). The best-fit linear regression lines and 95% confidence intervals are shown. Each point represents the mean of duplicate measurements for an individual. *$p < 0.05$, **$p < 0.01$. Source data are provided as a Source data file.

in the disease[35,51] are linked: downregulation of *PICALM* lowering efflux of Aβ and downregulation of angiogenic VEGF-VEGFR, TGFβ and semaphorin signalling pathways reducing the functional vascular density[74,75]. Pathways involved in angiogenesis were enriched in the two-layer neighbourhoods of other GWAS genes that showed altered expression in EC, such as *MEF2C, CCN2* and *ADAM10*. Functionally less well characterised AD GWAS genes, *WWOX* and *IQCK*, have large neighbourhoods in the FB and PC co-expression networks associated with enrichment for pathways supporting maintenance of BBB integrity[9]. BBB dysfunction allows neurotoxic plasma proteins to diffuse into the brain extracellular space and is associated with cognitive decline[12].

Innate immune responses are central to AD pathogenesis and progression but have not been well defined in the brain microvasculature to date[76,77]. We found evidence for cell-specific differences in vascular inflammatory responses to AD with an upregulation of innate immune response genes, in PC and FB, in particular. In EC, we showed a downregulation of IFN signalling genes in EC. Perhaps surprisingly, INFγ and IL6-related signalling genes (and downstream protective *STAT3*[78–80]) also were downregulated in AD in EC. This downregulation could represent a partially adaptive response to chronically enhanced TLR signalling with AD[81–83]. PC also appear to play a role in vascular inflammatory mediation of early AD. Recently identified risk genes *CD46*, encoding a serine protease which mediates inactivation of complement proteins, and *IRAK3*, encoding a homeostatic mediator of innate immune responses[84], were upregulated and downregulated, respectively, in PC.

The most strikingly differentially expressed gene sets in microvascular cells with AD are involved in angiogenesis and vascular homoeostasis. VEGF/VEGFR and insulin signalling pathways[85] in EC and EGF/EGFR signalling in EC and PC were downregulated with AD[86] despite upregulation of other genes (e.g., *HIF1A, ANGPT2* and *FGF2)* associated with pro-angiogenic regulation[87]. These results add to prior evidence of dysfunctional angiogenesis in AD[18,70,88]. We have extended descriptions by showing that, despite upstream angiogenic signals (e.g., upregulation of *HIF1A*) and metabolic adaptations, major downstream effector pathways fail to respond at the transcriptional level. All of the aforementioned trophic pathways converge in EC on the Ras signalling pathway and genes central to this pathway (e.g., *RAC1, RASAL2, SPTBN1*) were significantly downregulated in our dataset. This result was reproduced in another vascular snRNAseq[17] and a large bulk RNAseq generated with integration of data presented from AD and control tissues in earlier studies[42–44]. Downregulation of Ras signalling appears to have a central role in the angiogenic and BBB dysfunction[37,89–91]. Our data suggest four different mechanisms could be responsible for this: (i) a decrease in VEGFA production in the vasculature by PVM and astrocytes[92]; (ii) increased expression of ANGPT2, which, although it can stimulate angiogenesis, also can increase vascular leak and instability when expressed to high concentrations by inhibiting the Ang1-TIE2 pathway[31–33]; (iii) proinflammatory signalling from PVM and astrocytes; and (iv) a direct proapoptotic effect and accelerated senescence with greater Aβ load.

PC appear to play a central role in the dysfunctional angiogenesis. Our data provide evidence for disruption of EGF/EGFR, PDGF and Wnt/β-catenin pathway signalling[93–95], which are involved in development

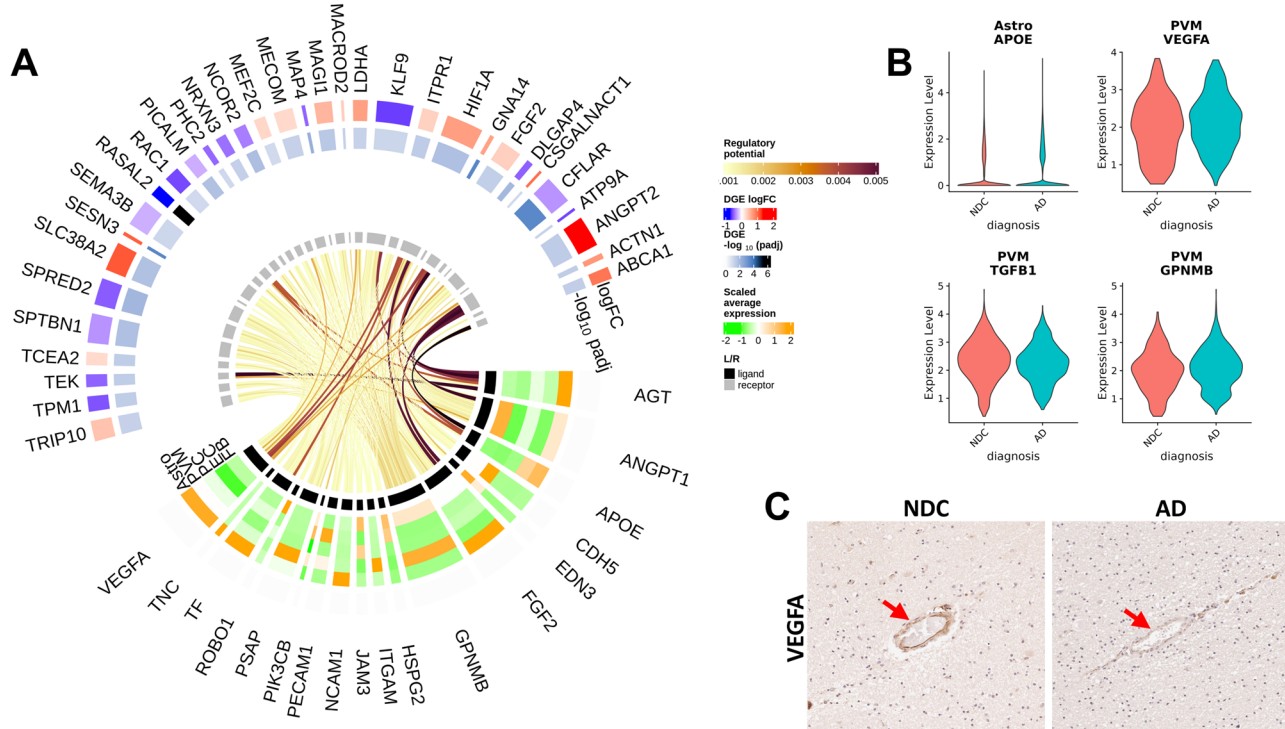

**Fig. 8 | NicheNet intercellular communication analysis identified potential regulators of EC DEG associated with proinflammatory and anti-angiogenic gene expression in astrocytes and perivascular macrophages. A** Circular heatmap and chord diagram of the results of the NicheNet analysis. The circular plot is divided (based on the innermost track) to separately represented ligand (black track) and target genes (grey track). A heatmap for the ligand genes on the 2nd to the 6th tracks (from inner- to outer-most) represents the average scaled expression of each ligand in each of the "sender" vascular cell types (each cell type -$\log_{10}$(padj) is represented on a different track). The two outermost tracks describe the differential expression (logFC) of the associated genes in the "received" EC. The outermost track represented the logFC and the 2nd outermost track represents the value. The links of the diagram represents the regulatory potential between the ligand and the target genes. **B** Astrocytic *APOE* (logFC=0.96, pval = .03), PVM *VEGFA*

(logFC = −0.31, pval=0.08), *TGFB1* (logFC = −0.28, pval=0.0007) and *GPNMB* (logFC=0.23, pval=0.04) potentially regulate the majority of EC DEG and are significantly differentially expressed with AD relative to NDC. Statistical significance was determined using a likelihood ratio test with a mixed-effects model and a zero-inflated negative binomial distribution. The analysis was performed on 170'299 astrocytic and 14'861 PVM nuclei from 57 independent samples. *P* values (unadjusted) refer to two-sided statistical tests. **C** IHC in a sample form the entorhinal cortex of sections from NDC (left) and AD (right) donors binding of VEGFA in the vessel wall with AD. Arrowheads denote the VEGFA binding in the vascular wall. The IHC experiment was performed on 24 independent samples. Scale bar = 50 μm. Source data are provided as a Source data file.

and maintenance of the BBB[35,96] and in the recruitment of PC for angiogenesis[97,98]. Our finding of *ADAM10* upregulation in PC also suggests an additional mechanism contributing to BBB breakdown, as ADAM10 promotes soluble PDGFRβ shedding associated with pericyte degeneration[99]. Increased soluble PDGFRβ in the CSF correlates with elevated CSF albumin and pTau and with cognitive impairment[14].

There are limitations to our analyses which should be addressed in future work. Our data was generated from nuclei from multiple brain regions and thus could address robustly only those transcriptomic differences common to the different regions. Regional variations in reductions of cerebral blood flow with early AD[100] highlight the potential for regional differences in some molecular mechanisms. A second limitation was that we assessed the total extracted populations of nuclei without seeking to identify and separately study cells expressed from specific vascular anatomic zones[17,19]. Nevertheless, the high overlap of our cellular markers and the cellular markers from human and mouse brain vessel-associated cells in previous reports provides assurance that all major cell types were represented. Third, like other recent studies, our conclusions are based on relatively sparse (10X Genomics Single Cell 3' Gene Expression assay) sequencing of the nuclear transcriptome, which may be biased relative to those from the whole cell, potentially reducing the power to detect transcripts from some genes[101]. Use of larger numbers of nuclear and

co-expression-based analyses, which rely less on detection of absolute expression levels than do single gene differential expression analyses, mitigates this limitation to some extent, but future confirmation will require more comprehensive transcriptional analyses of the whole cells.

In summary, our work has described quantitatively significant EC enrichment in AD risk genes suggesting that their specific contribution to inflammatory activation and reduced Aβ clearance are early, potentially "causal" factors in the onset of sporadic late onset AD. Evidence for impairment of angiogenesis and vascular homoeostasis with greater Aβ and ptau describes how microvascular molecular pathology arises that can potentiate brain hypoxia and proinflammatory activation and reduce endothelial Aβ clearance with AD. Exploration of the expression of risk genes for AD suggests additional mechanisms by which this could occur, such as through progressively greater reductions in EC expression of *PICALM*, as Aβ levels rise[77]. The extraordinary length of the brain capillary network (-650 km) and its large surface area (-120 cm²/g) suggest that even small relative effects could contribute substantially to increasing the overall Aβ burden in the CNS[33]. More generally, our results suggest that therapeutic targets in EC related to angiogenic, inflammatory and Aβ clearance pathways deserve prioritisation in the search for treatments able to slow or prevent the onset of AD.

## Methods

Data for this study was generated from (i) cortical brain tissue processed locally as we described earlier[22], (ii) from that described and made available publicly previously[21] and (iii) from brain tissue that was dedicated to this study.

### Local tissue access and snRNA sequencing

**Brain cortical tissue.** The experiments at Imperial College London were performed in accordance with National Research Ethics Committee approvals through the UK Brain Banks providing them and approvals of the UK Human Tissue Authority with governance oversight from Imperial College London. All of the brain banks in the UK Brain Bank Network have generic ethics committee approval to function as research tissue banks, which means that they can provide tissue samples to UK-based researchers for a broad range of studies without the need for the researchers to obtain their own ethics approval. (https://brainbanknetwork.ac.uk/public/en/researchers/accessingtissue/#:-:text=All%20of%20the%20brain%20banks,obtain%20their%20own%20ethics%20approval.) Part of the samples were obtained from the Oxford Brain Bank under ethics committee approval number 15/SC/0639. Ethics oversight of the experiments performed by ref. [21] are described in the published manuscript. Tissues and processing were described previously[22]. Cases were selected from the London Neurodegeneration (King's College London), Parkinson's UK (Imperial College London), Oxford, Edinburgh and South West Dementia Brain Banks. For snRNAseq and IHC experiments, entorhinal and somatosensory cortex from 6 non-diseased control (NDC) cases (Braak stage 0–II) and 6 AD cases (Braak stage III–VI) (total of 24 brain samples) and prefrontal cortex samples from 10 NDC and 10 AD cases (total of 20 samples) were used (Table 1). The total number of locally processed samples for snRNAseq and IHC was 44. In addition, we employed 10 temporal cortex samples from 4 NDC and 6 AD cases for ELISA experiments (Table 2). Brains used for this study excluded cases with clinical or pathological evidence for small vessel disease, stroke, cerebral amyloid angiopathy, diabetes, Lewy body pathology (TDP-43), or other neurological diseases. Where the information was available, cases were selected for a brain pH greater than 6 and all but one had a *post mortem* delay of less than 24 h.

### Immunohistochemistry

Immunohistochemical staining was performed on formalin-fixed paraffin-embedded sections from contralateral regions of each sample used for snRNASeq from the entorhinal, somatosensory and prefrontal cortex. Standard immunohistochemical procedures were followed using the ImmPRESS Polymer (Vector Laboratories) and Super-Sensitive Polymer-HRP (Biogenex) kits (Supplementary file 10). Briefly, endogenous peroxidase activity and non-specific binding were blocked with 0.3% $H_2O_2$ and 10% normal horse serum, respectively. Primary antibodies were incubated overnight at 4 °C. Species-specific ImmPRESS or SuperSensitive kits and DAB were used for antibody visualisation. Counter-staining for nuclei was performed by incubating tissue sections in haematoxylin (TCS Biosciences) for 2 min. AD pathology was assessed by Aβ (4G8, BioLegend 17−24) and pTau (AT8, NBS Biologics) immunolabelling. To confirm antibody specificity, for each target, the exact same staining protocol was applied without the addition of the primary antibody (Fig. S13).

Labelled tissue sections were imaged using a Leica Aperio AT2 Brightfield Scanner (Leica Biosystems). Images were analysed using HALO software (Indica Labs, Version 2.3.2989.34). The area quantification and multiplex image analysis macros were used for the study.

### Nuclei isolation and enrichment for lower abundancy cell populations.
Local processing of the fresh frozen entorhinal and somatosensory cortical tissue blocks was performed as for our previous work[22]: we began with sectioning to 80 µm on a cryostat and grey matter separated by scoring the tissue with sharp forceps to collect ~200 mg grey matter in an RNAse-free Eppendorf tube. Nuclei from NDC and AD samples then were isolated in parallel using a protocol based on ref. [102]. All steps were carried out on ice or at 4 °C. Tissue was homogenised in a 2 ml glass douncer containing homogenisation buffer (0.1% Triton-X + 0.4 µ/µl RNAseIn + 0.2 µ/µl SUPERaseIn). The tissue homogenate was centrifuged at 1000 g for 8 min, and the majority of supernatant removed without disturbing the tissue pellet. Homogenised tissue was filtered through a 70 µm filter and centrifuged in an Optiprep (Sigma) density gradient at 13,000 g for 40 min to remove myelin and cellular debris. The nuclei pellet was washed and filtered twice in PBS buffer (PBS + 1% BSA + 0.2 µ/ml RNAseIn). Isolated nuclei were labelled in suspension in 1 ml PBS buffer with 1:500 anti-NeuN antibody (Millipore, MAB377, mouse) and 1:250 anti-Sox10 antibody (R&D, AF2864, goat) for 1 h on ice. Nuclei were washed twice with PBS buffer and centrifuged at 500 g for 5 min. Nuclei were incubated with Alexa-fluor secondary antibodies at 1:1000 (goat-anti-mouse-647 and donkey-anti-goat-488) and Dapi (1:1000) for 30 min on ice, and washed twice. Nuclei were FACS-sorted on a BD Aria II, using BD FACSDiva software (v9.0), gating first for Dapi +ve nuclei, then singlets and then Sox10- and NeuN-negative nuclei. A minimum of 150,000 double-negative nuclei were collected.

We also isolated nuclei, that were from tissue immediately adjacent to the samples described above but not subjected to the FACS-enrichment step (total brain nuclei dataset) – rather, directly processed for single-nucleus capture and snRNA sequencing as described in the following section. In this way, we obtained an unbiased representation of all the brain cell types. The resulting dataset was used for the analysis described in the "Enrichment of brain cell types in AD and WMH GWAS signal" section.

### Microvessel single nuclei isolation from fresh frozen human prefrontal cortex using a dextran gradient-based approach.
Samples from the prefrontal cortex were processed for microvessel single nuclei isolation. All steps were carried out on ice or at 4 °C. Tissue was processed to first isolate microvessels and then to release single nuclei for sequencing. Tissue was dounce homogenised and then centrifuged at 1000 g for 3 min. Pellets were re-suspended in 5.3% dextran and overlayed on a pre-prepared dextran gradient (12%, 16%). After centrifugation the bottom layer was subject to a second dextran gradient spin (8%, 20%), 4200 g for 20 min. The pellets were resuspended and passed through a 100 µm filter and then a 40 µm filter. The 40 µm filter was inverted, vessels washed off, and the resulting suspension centrifuged at 800 g for 8 min. The isolated microvessels were incubated with Collagenase 2 for 20 min and nuclei released by grinding with a pestle and filtering through a 20 µm filter.

### 10X Chromium barcoding and sequencing.
Sorted nuclei were centrifuged at 500 g, resuspended in 50 µl PBS buffer and counted in a LUNA-FL Dual Fluorescence Cell Counter (Logos Biosystems, L20001) using Acridine Orange dye to stain nuclei. Sufficient nuclei were added

**Table 1 | Cohort information for locally processed samples used in the snRNAseq and IHC experiments**

|  | Sex (F/M, number) | Age at death (yrs, mean ± SD) | Post mortem delay (hr, mean ± SD) | RIN (mean ± SD) |
|---|---|---|---|---|
| Non-diseased controls (Braak 0–II) | 25/16 | 80.61 ± 6.3 | 18.0 ± 6.9 | 4.9 ± 2.0 |
| Alzheimer's disease (Braak III–VI) | 15/21 | 78.47 ± 9.3 | 22.1 ± 15.9 | 7.1 ± 0.7 |

**Table 2 | Cohort information for locally processed samples used in the ELISA experiments**

| | Sex (F/M, number) | Age at death (yrs, mean ± SD) | Post mortem delay (hr, mean ± SD) |
|---|---|---|---|
| Non-diseased controls (Braak 0–II) | 2/2 | 86.75 ± 8.3 | 45.38 ± 18.5 |
| Alzheimer's disease (Braak III–VI) | 3/3 | 78.66 ± 9.6 | 39.50 ± 16 |

for a target of 7000 nuclei for each library prepared. Barcoding, cDNA synthesis and library preparation were performed using 10X Genomics Single Cell 3′ Gene Expression kit v3 with 8 cycles of cDNA amplification, after which up to 25 ng of cDNA was taken through to the fragmentation reaction and a final indexing PCR was carried out to 14 cycles. cDNA concentrations were measured using Qubit dsDNA HS Assay Kit (ThermoFisher, Q32851), and cDNA and library preparations were assessed using the Bioanalyzer High-Sensitivity DNA Kit (Agilent, 5067-4627). Samples were pooled to equimolar concentrations and the pool sequenced across 24 lanes of an Illumina HiSeq 4000 according to the standard 10X Genomics protocol.

**Single nuclei RNA sequence analyses**

**Processing of FASTQ files.** Locally generated snRNASeq data were pre-processed using 10X Genomics Cell Ranger. Illumina sequencing files were aligned to the genomic sequence (introns and exons) using GRCh38 annotation in Cell Ranger v3.1. Nuclei were identified above background by the Cell Ranger software.

**Quality control, dataset integration, dimension reduction and clustering.** Feature-barcode matrices from CellRanger produced corresponding to the local dataset produced as described above were jointly processed with the feature-barcode matrices from a previously published dataset[21]. These were downloaded from the Gene Expression Omnibus (accession number GSE148822, https://www.ncbi.nlm.nih.gov/geo/query/acc.cgi?acc=GSE148822). Together, the two datasets were generated from 57 brain samples. Quality control (QC), dataset integration, dimension reduction and clustering were performed using the Nextflow pipeline nf-core/scflow (v0.7.1)[103].

QC was performed separately on each sample. Nuclei that had fewer than 200 features were excluded, whereas for the higher feature filtering criterion, an adaptive threshold was estimated in each sample, which was four median absolute deviations above the median feature number in the sample. Nuclei with more than 5% of mitochondrial gene counts were also excluded. Only genes that had at least one count in 5 nuclei per sample were retained. The QC also included an ambient RNA profiling using the DropletUtils package (v1.12.1)[104] using default parameters. Finally, multiple identification was performed with DoubletFinder (v2.0.3)[105] using 10 principal components based on the 2000 most variable features and a pK value of 0.005.

Sample integration was performed using the Liger package (v1.0.0)[23] incorporated in the nf-core/scflow pipeline (v0.7.1)[103]. The k value was optimised at 20 and the lambda value at 5. 3000 genes were employed in the integration process. The integration threshold was 0.0001 and the maximum number of performed iterations was set to 100. We used the Liger factors as input for dimensionality reduction through UMAP[24]. For clustering, we have used the low dimensional embedded output from UMAP for subsequent modularity optimisation and clustering using Leiden algorithm and a resolution parameter of 0.00001 and a k value of 50.

Cell-type identification of clusters was performed by identifying cluster-specific genes plotting canonical cell markers using the FeaturePlot function in Seurat (v3.2.3)[106]. To separate the vascular mural cells efficiently, we isolated the EC and the vascular mural cell clusters and re-ran the steps of the integration, dimension reduction and clustering. Cluster-specific genes were identified using the FindMarkers function in Seurat (using the MAST method[30] with the function arguments set to default). To validate the cell-type specificity of the

clusters and their identity, we compared the top 100 cluster markers of our dataset with the top 100 cluster markers of the same cell types from previously published human and mouse datasets[19,25] using an overrepresentation analysis.

**Overrepresentation analysis.** Overrepresentation analysis was performed to determine if the overlap between two gene sets was significantly higher that might occur by chance. This analysis was done using with the "enrichment" function of the R package bc3net (v1.0.4) (https://github.com/cran/bc3net), which performs a Fisher's exact test (FET). The $p$ values associated with the Fisher's exact test correspond to the probability that the overlap between the two gene sets and has occurred by chance.

**Enrichment of brain cell types in AD and WMH GWAS signal.** GWAS summary statistics for AD[3] and WMH (a radiological manifestation of small vessel disease)[29] were tested for enrichment in brain cell types using the MAGMA.Celltyping (v1.0.1)[5,107] and MungeSumstats (v1.1.24)[108] packages. First, summary statistics were appropriately formatted using MungeSumstats for use with MAGMA.Celltyping. Then, SNP associations from the summary statistics were mapped to genes using the map.snps.to.genes function of MAGMA.Celltyping. Next, as described for the default workflow of MAGMA.Celltyping, genes with low variability between the cell clusters were dropped using the drop_uninformative_genes and then quantile groups for each cell type were prepared using the prepare.quantile.groups function.

We first calculated the enrichment in AD and WMH GWAS signal across all the brain cell types on the dataset that had not been subjected to the FACS enrichment step to remove neurons and oligodendrocytes (see "Nuclei isolation and enrichment for lower abundancy cell populations" section). This was performed using the calculate_celltype_associations function with default parameters and the "linear" enrichment mode. This analysis was repeated after controlling for the microglial enrichment of the GWAS signal. Next, we calculated the enrichment in AD and WMH GWAS signal on each of the vasculature-associated cell types (EC, FB, PC and SMC). Finally, we assessed if the enrichment of the vasculature-associated cell types in our dataset in AD GWAS signal changed after controlling for the enrichment in WMH GWAS signal. For this, we re-ran the calculate_celltype_associations function for the AD summary statistics and the SNP-to-gene mapping of the WMH GWAS (that was calculated earlier with the map.snps.to.genes function) in the genesOutCOND argument of the function.

**Differential gene expression analysis.** DGE analysis was performed using MAST (v1.18.0). The transcriptomic alterations in AD vs NDC samples were assessed separately in each cell type by means of a zero-inflated negative binomial regression analysis by fitting a mixed-effects model. The use of a mixed-effects model is particularly important in the context of snRNAseq DGE analyses to account for the pseudoreplication bias that would otherwise be observed if a fixed-effects-only model was employed[109]. The model specification was zlm(~diagnosis + (1|sample) + cngeneson + pc_mito + sex + brain_region, sca, method = "glmer", ebayes = F). The fixed effect term pc_mito accounts for the percentage of counts mapping to mitochondrial genes. The term cngeneson is the cellular detection rate. Each nuclei preparation was considered as a distinct sample for the mixed effect. Models were fit with and without the independent variable and compared using a

likelihood ratio test. Units for differential expression are defined as log2 fold difference in AD vs NDC nuclei. The inclusion of a "dataset" term in the model was not necessary because the inclusion of the brain region term completely accounted for it. In the subset of samples that corresponded to the dataset produced in our laboratory, we also performed an exploratory regression analysis of gene expression against the two abovementioned histopathological features (using pTau or Aβ as markers) using MAST. The model specification was zlm(~histopath_marker + (1|sample) + cngeneson + pc_mito + sex + brain_region, sca, method = "glmer", ebayes = F). In this case, units for differential expression are defined as log2 fold difference/% pTau positive cells (or log2 fold difference/% Aβ area), i.e., a one-unit change in immunohistochemically-defined pTau (or Aβ) area density is associated with one log2-fold change in gene expression. In both MAST analyses, genes expressed in at least 10% of nuclei from each cell type were tested. Genes with an adjusted $p < 0.1$ were defined as meaningfully differentially expressed.

**Gene ontology and pathway enrichment analysis.** The gene ontology (GO) enrichment and the pathway enrichments analysis were carried out using the R package enrichR (v 3.0), which uses FET (Benjamini-Hochberg FDR < 0.1)[110]. Gene sets with minimum and maximum genes of 10 and 500 respectively were considered. To improve biological interpretation of functionally related gene ontology and pathway terms and to reduce the number of redundant gene sets, we first calculated a pairwise distance matrix using Cohen's kappa statistics based on the overlapping genes between the enriched terms and then performed hierarchical clustering of the enriched terms[111].

**Gene co-expression analysis.** Gene co-expression modules and hubgenes were identified separately for each cell type using the MEGENA (v1.3.7) package[36]. MEGENA constructs a hierarchy of co-expression modules with larger ("parent") that are further divided into subset ("children") modules. "Children" modules are subsets of the "parent" ones and have higher numbers as names than their "parents". To reduce the effect of noise, due to the sparsity of the expression matrix in a snRNAseq experiment, a sample-level pseudo-bulking was performed by summing the raw counts of all the nuclei in a sample. Genes expressed in at least 50% of the samples were used as input. The MEGENA pipeline then was applied using default parameters, using Pearson's correlations and a minimum module size of 10 genes. Parent modules were produced from which a sub-set of genes form smaller child modules (Supplementary file 5). The co-expression module hierarchy was represented graphically using Cytoscape software (Mac OS version 3.8.0)[112].

**Expression-Weighted Cell type Enrichment (EWCE).** EWCE (v1.4.0)[113] assesses if a user-provided gene set is associated, in terms of gene expression, to any particular cluster in a single cell/nuclei RNAseq dataset. It calculates the significance of the overlap between the gene set and the genes that are representative of each cluster using a bootstrapping approach. In this paper, we used two gene sets that are characteristic of microglia[27] and astrocytes (the top 100 genes associated to astrocytes from[28]) and calculated their enrichment in our dataset using default parameters.

**Cell-cell communication analysis.** Cell-cell communication analysis was performed using NicheNet (v1.1.1)[69]. NicheNet uses gene expression information from the dataset under study and integrates this information with a prior model of built by integration prior knowledge on ligand-to-target signalling paths. NicheNet seeks to identify ligands expressed in "sender" cell types that can regulate a user-provided set of genes in the "receiver" cell type, not limited to cognate receptors of these ligands, but any potential downstream gene target. The NicheNet algorithm with default parameters (unless otherwise specified) was applied to the part of our dataset that included all vascular cell types and we also included PVM and astrocytes to assess their potential involvement in regulating transcription in vascular cells. The analysis was performed jointly on the AD and NDC samples. The target set of genes in "receiver" cells were the significantly up- and downregulated genes in the respective cell type. Our objective was to prioritise ligand genes in "sender" cells that may potentially drive the differential expression in each "receiver" cell type. Genes that were expressed in at least 5% of nuclei were included in the analysis. The results were plotted on heatmaps that show the regulatory potential of the potential ligand genes in "sender" cell types (vertical axis) and the DEG-target genes in the "receiver" cell type (horizontal axis).

**Aβ42 enzyme linked immunosorbent assay (ELISA).** 11 temporal cortex samples were employed in the ELISA experiments. Guanidine-extracted insoluble fractions were prepared as reported previously[114]. A commercial sandwich ELISA (R&D Systems) was used according to manufacturer's instructions to measure amyloid-$\beta_{42}$. Guanidine extracted samples were diluted 1:2500 for AD samples and 1:625 for NDC. Each sample was measured in duplicate and the means calculated after interpolation from serial dilutions of recombinant human amyloid-$\beta_{42}$ (7.8–500 pg/ml) corrected for sample dilution.

**Myelin-associated glycoprotein, proteolipid protein-1, CD31, CD105 and fibrinogen enzyme linked immunosorbent assays.** Fresh frozen brain tissue was diluted 20% w/v in 1% sodium dodecyl sulphate (SDS) lysis buffer [1% w/v SDS, 0.1 M sodium chloride (NaCl), 0.01 M Tris hydrochloride (Tris-HCl) (pH 7.6), 1 μg/ml of aprotinin and 1 μM phenylmethylsulphonyl fluoride (PMSF)] and homogenised with 5–10 silica beads (2.3 mm diameter) in a Precellys homogenizer (2 × 15 s at 6000 rpm). Homogenates were aliquoted and stored at −80 °C prior to use. The concentration of MAG was determined by in-house direct ELISA[16,115,116] (Abcam, diluted 1:1000 in PBS) and the concentration of PLP1, fibrinogen, CD31 and CD105 by sandwich ELISA (PLP1, SEA417Hu; Cloud-Clone Corp., USA/China, fibrinogen EH3057; FineTest, China, CD31, DY806-05; R&D systems, USA, CD105, KE00199; Proteintech, USA), methods identical to those used in our previous work[15,16,80,116–118].

**Reporting summary**
Further information on research design is available in the Nature Portfolio Reporting Summary linked to this article.

## Data availability
The snRNAseq data are available for download from the Gene Expression Omnibus (GEO) database (https://www.ncbi.nlm.nih.gov/geo/) under accession number GSE160936 and GSE252921. Previously described data[21] was downloaded from the GEO database (GSE148822). Source data are provided with this paper.

## Code availability
Analysis scripts used in this manuscript are available on GitHub (https://github.com/stergiostsartsalis/A-single-nuclear-transcriptomic-characterisation-of-mechanisms-responsible-for-impaired-angiogenesis/tree/1)[119].

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

## Acknowledgements

We thank the donors and their families for the use of human brain tissue in this study and staff of the London Neurodegeneration, Oxford, Edinburgh, Parkinson's UK and Southwest Dementia Brain Banks for making it available. Infrastructure, including the Imperial College British Heart Foundation Centre of Excellence, the LMS/NIHR Imperial Flow Cytometry Facility and the Imperial College Genomics Facility, which are supported by the National Institute for Health Research (NIHR) Biomedical Research Centre (BRC), all contributed to the work. ST was supported by an "Early Postdoc.Mobility" scholarship (P2GEP3_191446) from the Swiss National Science Foundation, a "Clinical Medicine Plus" scholarship from the Prof Dr Max Cloëtta Foundation (Zurich, Switzerland) and a scholarship from the University Hospitals of Geneva. P.M.M. acknowledges generous personal support from the Edmond J Safra Foundation and Lily Safra and an NIHR Senior Investigator Award. This work additionally was supported by awards to P.M.M., A.M. and C.W. from the UK Dementia Research Institute, which receives its funding from UK DRI Ltd., funded by the UK Medical Research Council, Alzheimer's Society and Alzheimer's Research UK. This project has received funding (ZC) from the Innovative Medicine Initiative 2 Joint Undertaking under grant agreement No 807015. This Joint Undertaking receives support from the European Union's Horizon 2020 research and innovation programme EFPIA. The manuscript reflects the authors' view and that neither IMI nor the European Union, EFPIA, or any Associated Partners are responsible for any use that may be made of the information contained therein. Part of this work was performed using the computational facilities of the Advanced Research Computing @ Cardiff (ARCCA) Division, Cardiff University.

## Author contributions

S.T.–experimental design, primary data analysis, interpretation of results, draughted the manuscript. H.S. – experimental design, performed experiments, primary data analysis. N.F. – experimental design, primary data analysis, interpretation of results. F.W. – performed experiments, primary data analysis. A.M.S.- performed experiments, data analysis, interpretation of results. N.W. – performed immunohistochemistry experiments and image analysis. T.K.D.C. – performed immunohistochemistry experiments and image analysis. M.J.R. – performed experiments, primary data analysis. V.C. – performed experiments. E.I. – performed immunohistochemistry experiments and image analysis. C.K. – software developments, support for data analysis. O.A. – experimental design, interpretation of results. X.Y. – primary data analyses, interpretation of results. M.H.J. – primary data analyses, interpretation of results. K.D. – performed experiments, interpretation of

results. A.Mc.- performed immunohistochemistry experiments and image analysis. R.C.J.M.- performed immunohistochemistry experiments and image analysis. S.D. – supported data analysis, interpretation of results. J.S.J. – experimental design, interpretation of results. A.M. – supported data analysis, interpretation of results. D.R.O. – experimental design, interpretation of results. J.S.M. – experimental design, performed experiments, primary data analyses, interpretation of results. S.L. – experimental design, interpretation of results. C.W. – experimental design, interpretation of results. M.Z.C. – experimental design, interpretation of results. P.M.M.- experimental design, interpretation of results, draughted the manuscript. All of the co-authors reviewed and approved the final manuscript.

## Competing interests

P.M.M. is a consultant for Biogen, Sudo Therapeutics, Nimbus, Astex, GSK and Sangamo. He has received research funding for aspects of this work from Biogen and the UK DRI. He has research funding unrelated to this work from Biogen and Bristol Meyers Squibb. ZC is founder and director of Oxford StemTech and Human-Centric DD. The remaining authors declare no competing interests.
