## [Peer Review File · Nature Communications]

A single nuclear transcriptomic characterisation of mechanisms responsible for impaired angiogenesis and blood-brain barrier function in Alzheimer's diseaseReviewers' Comments:

Reviewer #1 (Remarks to the Author):

Using FACS to enrich vascular cells for snRNAseq, Tsartsalis et al., report the transcriptomic characteristics of vascular cells in AD vs. non-AD control subjects. The main findings included (i) an EC-enrichment of AD GWAS susceptibility gene expression, (ii) complex expression changes implicating altered angiogenesis, immune signalling and homeostatic function in the vasculature in AD, and (iii) vascular expression changes associated with AD neuropathology.

The biggest contribution of this study to the field, are (A) further detailed insights into the transcriptional changes in vascular cells in AD and (B) the provision of a high-quality dataset for future data mining by other groups. The latter is just as important, as AD cohorts certainly exhibit variability from one to another. The dataset provided by the current study will also be an important one for testing what are consistent or inconsistent across cohorts.

I have the following comments which I hope would help to the strengthen the manuscript:

- If the authors want to highlight a certain list of DEGs with important functional implications, such as those highlighted in the text, at least please ensure they are annotated in the corresponding figures (and even better if they are highlighted in the plots and catch more attention on first sights too). For example, I could not find HIF1A in Fig 2A. Same applies to numerous other DEGs mentioned in the main text.

- Did the authors attempt to perform comparative analyses, using previously reported datasets (e.g., Grubman et al., Nature Neuroscience 2019; Zhou et al., Nature Medicine, 2020; and Lau et al., PNAS 2020)? Although these previous studies had less vascular cells, it would be good to have the results of comparative analysis included in this manuscript, as (i) it is important for the field to work out what are shared and what differ across studies / cohorts, and (ii) it actually helps the current manuscript, if the comparative analysis can reveal what could not be uncovered by the previously reported datasets.

- The authors chose to report the last part of the results in the form of supplementary tables, yet I personally think that the results are nice. Very few studies / datasets have the opportunity to examine what vascular differential expressions are associated with the severity of neuropathological changes (A-beta and tau loads here). I strongly suggest the authors to present this part in a figure instead. I understand that the pattern would appear complicated and it is often difficult to attribute causal vs. purely correlative non-causal relationships (as is the case for many transcriptomic findings), but nonetheless they may provided important new insights.

- Lines 115 and 603 - Based on my understanding, clustering with the Leiden method typically refers to clustering by building a kNN graph followed by modularity optimization using the Leiden (or other) algorithm. UMAP is a dimensionality reduction algorithm, also with a step of building a kNN graph. Did the authors meant to say that they used the kNN graph built during UMAP, or the low dimensional embedded outputs from UMAP for subsequent modularity optimization with the Leiden algorithm, when they mentioned that clustering was done using UMAP? Please revise the writing and clarify.

Reviewer #2 (Remarks to the Author):

In this article, the authors used FACS to remove glia cells and neurons for snRNA-seq to represent vascular cells better and then integrated their data with a published dataset to study the potential 'casual' effects on AD pathogenesis. Previous publications suggest that brain vascular cells play an important role in AD pathophysiology, so this paper emphasized analyzing single nuclear transcriptional signatures of dysfunctional brain vascular homeostasis in Alzheimer's disease. In the analysis, the authors focused on the enriched expression of AD risk genes in brain microvascular cells and the functional enrichment of co-expressed genes and pathways altered in AD vascular cells. The authors took GWAS summary statistics and snRNA-seq to find the association between Alzheimer's diseases and the cell types. They then argued that endothelial cells enriched in genes were associated with AD risk. Following that, the authors identified both potential DEGs and co-expression modules for maintaining the integrity of vascular cells. Furthermore, through cell-cell communication, authors wanted to identify ligand-receptor interaction shared between AD and control or only in AD or control to find the impaired signaling mechanism responsible for the vascular integrity.

However, several aspects raised my concerns regarding the study data and results. The authors focused a lot of analysis on showing the enrichment of GWAS variants in vascular cells and spent a significant portion of the analysis understanding the genes responsible for this GWAS enrichment. For example, in Figure 1D, the authors showed the 'relative' enrichment of vascular cells in GWAS variants using MAGMA. The authors concluded that the enrichment of vascular nuclei is reduced (not significant) after controlling for genes enriched in microglia (Fig. 1E). If the risk variants are not enriched in vascular cells, there is no need to look for AD-risk genes using 2-layered approach in Figure 4. The authors could have analyzed the data better to understand the biological processes that are altered during AD in relation to vasculature. Most of the analysis is very basic and preliminary and deeper insight into biology is warranted. The cell-chat analysis in Fig 5 is represented in a very preliminary way and further analysis is needed to understand the effect of cell-cell communication in AD biology. Discussion also does little to unravel what is novel in this analysis, at the end of the day, this is a very poor analysis of the data. Basic plots of UMAP to understand batch effects in the data, and quality of data integration with external datasets are also not provided, making it hard to evaluate the data quality.

Reviewer #3 (Remarks to the Author):

In this study, Tsartsalis et al. present single nuclear transcriptional analysis of vascular cells in Alzheimer's disease. They isolated endothelial cells (EC), pericytes (PC), fibroblasts (FB), and smooth muscle cells (SMC) from human entorhinal and somatosensory cortex, combined it with a published snRNA-Seq dataset, and their analysis suggests dysregulation of vascular homeostasis and angiogenesis, impaired β -amyloid clearance, and immune activation in these vascular cells. The analysis is rigorous, the methods are clearly written, and the work adds to significant work on the role of the brain microvasculature in Alzheimer's disease.

The main findings, however, are not novel: dysregulation of angiogenesis was demonstrated by Lau et al. and the potential involvement of AD risk genes (AD GWAS genes) in the human brain vasculature was introduced by Yang et al. The authors suggest that, among vascular cells, risk genes associated with AD are enriched most in EC, but immunohistochemical confirmation is not shown. The cell-cell communication analysis, which further supported the idea of impairment in vascular homeostasis and angiogenesis in AD, is new but additional follow-up experiments to validate these findings would strengthen the study.

In addition, there are a few other limitations. The number of nuclei for each cell type is not reported; it is simply mentioned that the number of SMC is small. Although lack of zonation analysis is mentioned as a limitation in the discussion, adding such analysis would be a plus.

The authors used 24 samples from entorhinal and somatosensory cortex (N=24 samples, 6 AD and 6 CTR donors) and combined it with a previously published dataset by Gerrits et al. (N=36 samples from 10 AD and 5 CTR and 3 CTR+ donors). It is unclear how that adds to the 57 samples (31 AD and 26 CTR) described in the results section, and whether the CTR+ donors (non-demented controls with mild amyloid- β pathology) were removed from the analysis. The other reason for concern is the integration across regions. Although the authors claim that "nuclei from different datasets and brain regions were well-mixed after integration", Figure S1C shows that not all clusters had equal representation from the four brain regions, suggesting potential region-specific changes. This issue is noted in the discussion though.

The authors perform both differential expression analysis using a mixed effects model and report transcriptional changes in angiogenesis, amyloid processing, and immune responses. They also identified gene co-expression modules differentially expressed with AD using MEGENA. They report similar results: reduced expression of co-expression modules enriched for angiogenesis and vascular homeostasis as well as changes in modules enriched in amyloid processing and immune response. Integrating these two approaches to focus on the functional modules and the key genes would make it much easier to digest the information. An interpretation of changes in the immune related modules in opposite directions is needed. A summary at the end of each results section would be also helpful for the reader.

The markers COL1A1, COL12A1, COL6A1 and COL5A1 used to identify fibroblasts are also expressed in SMC and pericytes (see Yang et al website). The authors state that they confirmed the cluster annotations with existing datasets (lines 130-133), but it would be helpful to confirm with Yang et al. as well since that is the most extensive study of brain vascular cells to date. In addition, to identify non-vascular cell clusters, authors used single markers [microglia (CD74), astrocytes (GFAP), oligodendrocytes (PLP1) and neurons (RBFOX3)]. However, GFAP for example may be present only in reactive astrocytes, so crosschecking with a few other markers would be helpful. Also, ensuring that the vascular cells are not contaminated by astrocytes or microglia RNA would be reassuring.

A few other minor errors:

89-92 awkward sentence: Proangiogenic HIF1A was overexpressed in EC in AD (Figure 2A), no HIF1A nor SPRED2, SHC2, KSR1, RASGRF2, DAB2IP, RASAL2, DUSP16, VCL and EGFR shown.

59-62: lengthy sentence with too many conjunctions.

311-312: Missing SORL1 for EC enrichment in Figure 4.

Typographical error in Figure S9: MAGMA.Celltyping enrichment of brain nuclei in genomic loci associated with genetic risk for WMH. Refers incorrectly to Figure S9 F.

Reviewers' comments on the original NCOMMS-21-41207 submission with Authors' responses

Reviewers' comments with Authors' responses in **BOLD** and new text in the manuscript in *Italics*

Reviewer #1 (Remarks to the Author):

Using FACS to enrich vascular cells for snRNAseq, Tsartsalis et al., report the transcriptomic characteristics of vascular cells in AD vs. non-AD control subjects. The main findings included (i) an EC-enrichment of AD GWAS susceptibility gene expression, (ii) complex expression changes implicating altered angiogenesis, immune signalling and homeostatic function in the vasculature in AD, and (iii) vascular expression changes associated with AD neuropathology.

The biggest contribution of this study to the field, are (A) further detailed insights into the transcriptional changes in vascular cells in AD and (B) the provision of a high-quality dataset for future data mining by other groups. The latter is just as important, as AD cohorts certainly exhibit variability from one to another. The dataset provided by the current study will also be an important one for testing what are consistent or inconsistent across cohorts.

I have the following comments which I hope would help to the strengthen the manuscript:

1 If the authors want to highlight a certain list of DEGs with important functional implications, such as those highlighted in the text, at least please ensure they are annotated in the corresponding figures (and even better if they are highlighted in the plots and catch more attention on first sights too). For example, I could not find HIF1A in Fig 2A. Same applies to numerous other DEGs mentioned in the main text.

We thank the reviewer for this comment, we now use violin plots to highlight DEGs that we want to highlight in describing vascular cell gene expression difference in AD (e.g., in Figure 2, 5 and 6)

2 Did the authors attempt to perform comparative analyses, using previously reported datasets (e.g., Grubman et al., Nature Neuroscience 2019; Zhou et al., Nature Medicine, 2020; and Lau et al., PNAS 2020)? Although these previous studies had less vascular cells, it would be good to have the results of comparative analysis included in this manuscript, as (i) it is important for the field to work out what are shared and what differ across studies / cohorts, and (ii) it actually helps the current manuscript, if the comparative analysis can reveal what could not be uncovered by the previously reported datasets.

We agree regarding the potential usefulness of this. We have focused on larger datasets, as the smaller datasets cannot be expected to high independent true positive rates. We have now performed comparative analyses with the dataset from the Yang et al paper¹ and we also performed a joint analysis of three large bulk RNAseq studies of human brain in AD²⁻⁴. The datasets were selected rather than those suggested by the reviewer because they were large enough to allow individual analyses of endothelial cells, pericytes, fibroblasts and vascular smooth muscle cells. We use fast gene set enrichment analysis (fgsea⁵) to identify if the functional pathways that are enriched in our endothelial DEG are also enriched in the DEG from these two studies (paragraph 5 of section "Transcriptional signatures of pathological angiogenesis and blood-brain barrier integrity with AD" in the Results).

3 The authors chose to report the last part of the results in the form of supplementary tables, yet I personally think that the results are nice. Very few studies / datasets have the opportunity to examine what vascular differential expressions are associated with the severity of neuropathological changes (A-beta and tau loads here). I strongly suggest the authors to present this part in a figure instead. I understand that the pattern would appear complicated and it is often difficult to attribute causal vs. purely correlative non-causal relationships (as is the case for many transcriptomic findings), but nonetheless they may provide important new insights.

We thank the reviewer for this suggestion and have added this result in a main figure (Figure 6).

4 Lines 115 and 603 - Based on my understanding, clustering with the Leiden method typically refers to clustering by building a kNN graph followed by modularity optimization using the Leiden (or other) algorithm. UMAP is a dimensionality reduction algorithm, also with a step of building a kNN graph. Did the authors meant to say that they used the kNN graph built during UMAP, or the low dimensional embedded outputs from UMAP for subsequent modularity optimization with the Leiden algorithm, when they mentioned that clustering was done using UMAP? Please revise the writing and clarify.

We have now revised the text to eliminate any potential ambiguity in the interpretation (paragraph “Quality control, dataset integration, dimension reduction and clustering” in the Methods section). Firstly, we integrated datasets using Liger, followed by the utilization of Liger factors as input for dimensionality reduction through UMAP. For clustering, we have used the low dimensional embedded output from UMAP for subsequent modularity optimisation and clustering using Leiden algorithm.

Reviewer #2 (Remarks to the Author):

In this article, the authors used FACS to remove glia cells and neurons for snRNA-seq to represent vascular cells better and then integrated their data with a published dataset to study the potential 'casual' effects on AD pathogenesis. Previous publications suggest that brain vascular cells play an important role in AD pathophysiology, so this paper emphasized analyzing single nuclear transcriptional signatures of dysfunctional brain vascular homeostasis in Alzheimer's disease.

In the analysis, the authors focused on the enriched expression of AD risk genes in brain microvascular cells and the functional enrichment of co-expressed genes and pathways altered in AD vascular cells. The authors took GWAS summary statistics and snRNA-seq to find the association between Alzheimer's diseases and the cell types. They then argued that endothelial cells enriched in genes were associated with AD risk. Following that, the authors identified both potential DEGs and co-expression modules for maintaining the integrity of vascular cells. Furthermore, through cell-cell communication, authors wanted to identify ligand-receptor interaction shared between AD and control or only in AD or control to find the impaired signalling mechanism responsible for the vascular integrity.

1. However, several aspects raised my concerns regarding the study data and results. The authors focused a lot of analysis on showing the enrichment of GWAS variants in vascular cells and spent a significant portion of the analysis understanding the genes responsible for this GWAS enrichment. For example, in Figure 1D, the authors showed the 'relative' enrichment of vascular cells in GWAS variants using MAGMA. The authors concluded that the enrichment of vascular nuclei is reduced (not significant) after controlling for genes enriched in microglia (Fig. 1E). If the risk variants are not enriched in vascular cells, there is no need to look for AD-risk genes using 2-layered approach in Figure 4.

We apologise for the confusion. The logic of this section needs clarification. The observation that controlling for microglial enrichment of the GWAS signal reduces the independent enrichment of EC means that similar common sets of genes related GWAS loci are enriched in EC and microglia. Both cell types show relative enrichment in GWAS signal. To clarify this issue, at the end of the 5th paragraph of the following section in the Results “Endothelial cells are enriched in genes associated with genetic risk for AD”, we now add:

Together, these results suggest that the molecular mechanisms associated with the genetic risk for AD are expressed in EC, but largely overlap with those that also confer risk expressed in microglia.

2. The authors could have analyzed the data better to understand the biological processes that are altered during AD in relation to vasculature. Most of the analysis is very basic and preliminary and deeper insight into biology is warranted.

To obtain a better understanding of the biological processes we substantially expanded our data and the range of analyses: (1) we increased the size of our datasets with the addition of additional biological samples in our EC dataset; (2) we performed immunohistochemistry to confirm differences in expression of hypothesise target genes; 3) we described the DGE and the co-expression analyses together to highlight biological processes identified in both analyses; (4) we enhanced our regression models by including data regarding APOE genotype; and, (5) we have performed ELISA on human *post mortem* tissue homogenates to define biochemical pathology related to the transcriptomic differences described. to obtain more insight on targets associated with dysfunctional angiogenesis. We believe that the revised manuscript is now is substantially more comprehensive.

3. The cell-chat analysis in Fig 5 is represented in a very preliminary way and further analysis is needed to understand the effect of cell-cell communication in AD biology.

We thank the reviewer for this comment. These concerns have prompted us to use NicheNet⁶ to assess the cell-cell communication patterns that are potentially regulating the differential gene expression in target cell-types. To better characterise differences in vascular cell-cell communication alterations with AD, we included data concerning perivascular macrophages and astrocytes, glial cells with a potentially important role in the cross-talk with the vasculature. These together allowed more confident and meaningful conclusions to be drawn.

4. Discussion also does little to unravel what is novel in this analysis, at the end of the day, this is a very poor analysis of the data. Basic plots of UMAP to understand batch effects in the data, and quality of data integration with external datasets are also not provided, making it hard to evaluate the data quality.

As described above, we have now considerably modified the structure of results and discussion to focus on major biological processes that our data suggests are altered in AD. We have performed comparisons with previously published datasets (see response to comment 2 of reviewer #1). The quality of data integration is shown in Figures S1 and S10, which make clear that we did not have major batch effects confounding interpretations after the data integration.

Reviewer #3 (Remarks to the Author):

1 In this study, Tsartsalis et al. present single nuclear transcriptional analysis of vascular cells in Alzheimer’s disease. They isolated endothelial cells (EC), pericytes (PC), fibroblasts (FB),

and smooth muscle cells (SMC) from human entorhinal and somatosensory cortex, combined it with a published snRNA-Seq dataset, and their analysis suggests dysregulation of vascular homeostasis and angiogenesis, impaired β -amyloid clearance, and immune activation in these vascular cells. The analysis is rigorous, the methods are clearly written, and the work adds to significant work on the role of the brain microvasculature in Alzheimer's disease.

The main findings, however, are not novel: dysregulation of angiogenesis was demonstrated by Lau et al. and the potential involvement of AD risk genes (AD GWAS genes) in the human brain vasculature was introduced by Yang et al. The authors suggest that, among vascular cells, risk genes associated with AD are enriched most in EC, but immunohistochemical confirmation is not shown.

We thank the reviewer for the positive comments. We believe that we have substantially extended previous work. Our results provide a specific list of target genes and identify the precise components of the biological pathways that are altered in vascular cells in AD. For instance, our results describe how, despite evidence for increased expression of pro-angiogenic effectors (e.g., FGF and ANGPT2), expression of downstream effector signalling pathways is reduced. We suggest that this downregulation is potentially related to a failure to upregulate VEGFA in the endothelial cells due to pathological glia-EC cross-talk. We have drawn a specific relationship between this pathology and β -amyloid. We also have extended transcriptomic descriptions with biochemical and immunohistochemical analyses for major target genes/proteins identified.

2 The cell-cell communication analysis, which further supported the idea of impairment in vascular homeostasis and angiogenesis in AD, is new but additional follow-up experiments to validate these findings would strengthen the study.

We agree with the reviewer. First, to perform a cell-cell communication analysis with relevance for the transcriptomic changes in the vascular cell types, we have replaced the analysis in the first version of our manuscript with a NicheNet analysis. NicheNet predicts interactions of ligands in sender cells with target genes (not limited to cognate receptors of these ligands but to any potential downstream gene) in receiving cells by integrating gene expression data with prior models of signalling and gene regulatory networks. In our case, the target genes were the differentially expressed genes in each cell type. We thus identified which cell-cell communication patterns could be upstream regulators of the differentially expressed genes. Second, we included perivascular macrophages and astrocytes in the NicheNet analysis, as these two cell types may have important general influences on AD pathophysiology and their physical proximity to vascular cells suggests the potential for modulatory cross talk. Finally, we performed immunohistochemistry and validated the altered binding of ligands that could regulate these genes (e.g., VEGFA, ANGPT2, FGF2).

3 In addition, there are a few other limitations. The number of nuclei for each cell type is not reported; it is simply mentioned that the number of SMC is small. Although lack of zonation analysis is mentioned as a limitation in the discussion, adding such analysis would be a plus.

We now report the number of nuclei for each cell type (end of 2nd paragraph of the results) and we have assessed the enrichment of zonation markers in our cell types (Figure S4).

4 The authors used 24 samples from entorhinal and somatosensory cortex (N=24 samples, 6 AD and 6 CTR donors) and combined it with a previously published dataset by Gerrits et al. (N=36 samples from 10 AD and 5 CTR and 3 CTR+ donors). It is unclear how that adds to the 57 samples (31 AD and 26 CTR) described in the results section, and whether the CTR+

donors (non-demented controls with mild amyloid- β pathology) were removed from the analysis.

We have performed a joint analysis of the Gerrits et al and our own dataset using scFlow. scFlow uses more stringent criteria for considering the quality of a sample that reported in the original paper. Thus, some of the samples from the original Gerrits et al dataset were removed after the initial quality control. We did not exclude their “control” brain samples from our analysis. As the reviewer will appreciate, some degree of amyloid- β pathology is common even in healthy donor brains with low Braak stages.

5 The other reason for concern is the integration across regions. Although the authors claim that “nuclei from different datasets and brain regions were well-mixed after integration”, Figure S1C shows that not all clusters had equal representation from the four brain regions, suggesting potential region-specific changes. This issue is noted in the discussion though.

Supplementary Figure S1 and S10 showing UMAPs stratified in different ways illustrates good integration across datasets and regions. What the reviewer describes in Figure S1C is not related to a region effect on the distribution of cells among clusters, but instead to differences in numbers of cell isolated. Our dataset included nuclei from the entorhinal and the somatosensory cortex whereas the Gerrits et al dataset from the occipital and occipitotemporal cortex. The Gerrits et al dataset had more nuclei per sample, which was most apparent in clusters with least numbers of nuclei. As illustrated in the stacked barplot below, except for the apparently lower proportion in the entorhinal cortex (EC), the relative numbers of nuclei for each cell was relatively consistent across regions, despite differences in isolation and enrichment for vascular cells used by the difference investigators.

The original representation used a Seurat function (the “raster” parameter) that is active by default when the dataset includes more than 100,000 nuclei. Instead of plotting the actual data, rasterisation will plot an approximate density in square grids to avoid overplotting. We now present the data, using the same parameters as in the first version of our paper, except that the “raster” parameter is inactive to allow more accurate comparisons across different cell clusters and datasets with different numbers of nuclei.

6 The authors perform both differential expression analysis using a mixed effects model and report transcriptional changes in angiogenesis, amyloid processing, and immune responses. They also identified gene co-expression modules differentially expressed with AD using MEGENA. They report similar results: reduced expression of co-expression modules enriched for angiogenesis and vascular homeostasis as well as changes in modules enriched in amyloid processing and immune response. Integrating these two approaches to focus on the functional modules and the key genes would make it much easier to digest the information.

We thank the reviewer and agree. We have restructured our results and now jointly (and we believe more clearly) present the results of DGE and co-expression analyses.

7 An interpretation of changes in the immune related modules in opposite directions in needed.

We agree that this is an important finding that needs to be discussed. We now dedicate a considerable part of the fourth paragraph of the discussion to this finding:

Innate immune responses are central to AD pathogenesis and progression but have not been well defined in the microvasculature to date^{7,8}. We found evidence for cell-specific differences in vascular inflammatory responses to AD with an upregulation of innate immune response genes, in PC and FB, in particular. In EC, we showed a downregulation of IFN signalling genes in EC. Perhaps surprisingly, INF γ and interleukin 6 (IL6)-related signalling genes (and downstream protective STAT3^{9,10 11}) also were downregulated in AD in EC. This downregulation could represent a partially adaptive response to chronically enhanced TLR signalling with AD^{12 13,14}. PC also appear to play a role in vascular inflammatory mediation of early AD. Recently identified risk genes CD46, encoding a serine protease which mediates inactivation of complement proteins, and IRAK3, encoding a homeostatic mediator of innate immune responses¹⁵, were upregulated and downregulated, respectively, in PC.

8 A summary at the end of each results section would be also helpful for the reader.

We now have a brief summary at the end of each section.

9 The markers COL1A1, COL12A1, COL6A1 and COL5A1 used to identify fibroblasts are also expressed in SMC and pericytes (see Yang et al website). The authors state that they confirmed the cluster annotations with existing datasets (lines 130-133), but it would be helpful to confirm with Yang et al. as well since that is the most extensive study of brain vascular cells to date.

We have added this analysis (Figures S6-S7). In addition, we now provide the cluster marker genes as a supplementary file.

10 In addition, to identify non-vascular cell clusters, authors used single markers [microglia (CD74), astrocytes (GFAP), oligodendrocytes (PLP1) and neurons (RBFOX3)]. However, GFAP for example may be present only in reactive astrocytes, so crosschecking with a few other markers would be helpful.

We have now expanded Figure S2 and employ more cell markers for identification of microglia and astrocytes and, in Figure S3, oligodendrocytes and neurons. In addition, we now provide the cluster marker genes as a supplementary file. In the 3rd paragraph of the “Quality control, dataset integration, dimension reduction and clustering” section, we briefly describe how cell type identification was performed based on the full of co-expressed genes that distinguish each cluster.

11 Also, ensuring that the vascular cells are not contaminated by astrocytes or microglia RNA would be reassuring.

To verify this, we have employed Expression-Weighted Cell type Enrichment (EWCE)¹⁶. EWCE quantitatively assesses the expression of a user-defined gene set in independently defined, larger clusters of transcripts. It calculates the significance of the overlap between the gene set and the genes that are representative of each cluster using a bootstrapping approach. In this paper, we used two gene sets defining microglia¹⁷ and astrocytes¹⁸ and calculated their enrichments across the different nuclei transcriptional clusters in our dataset. This analysis showed that vascular cell clusters were not enriched significantly in microglia- or astrocyte-specific genes. We conclude that there is no significant contamination of any of our vascular cell clusters by transcripts from microglial or astrocytic nuclei. In the 5th paragraph of the following section in the Results “Endothelial cells are enriched in genes associated with genetic risk for AD”:

To exclude any significant contamination of the vascular cell clusters by microglia or astrocytes, we assessed their enrichment in gene expression sets that specifically characterize microglia and astrocytes using Expression-Weighted Cell type Enrichment (EWCE). The microglia- and astrocyte-specific gene sets were uniquely and significantly enriched in the corresponding cell clusters, whereas the clusters corresponding to vascular nuclei showed no enrichment (Figure S8).

Figure S8. EWCE enrichment of microglia- and astrocyte-specific gene sets. Vascular clusters show no enrichment of (A) microglia or (B) astrocyte-specific gene sets, confirming the purity of vascular clusters in our dataset. The parameter associated to the significance of the enrichment of each cell type is on the vertical axis. It corresponds to the difference, in terms of standard deviations, between the actual enrichment of each cell type (horizontal axis) and the mean enrichment of each cell type calculated using a bootstrapping approach (i.e., the enrichment that would be attributed to chance).

8 A few other minor errors:

89-92 awkward sentence: Proangiogenic HIF1A was overexpressed in EC in AD (Figure 2A), no HIF1A nor SPRED2, SHC2, KSR1, RASGRF2, DAB2IP, RASAL2, DUSP16, VCL and EGFR shown.

59-62: lengthy sentence with too many conjunctions.

311-312: Missing SORL1 for EC enrichment in Figure 4.

Typographical error in Figure S9: MAGMA.Celltyping enrichment of brain nuclei in genomic

loci associated with genetic risk for WMH. Refers incorrectly to Figure S9 F.

We have now rectified these minor errors

References

- 1 Yang, A. C. *et al.* A human brain vascular atlas reveals diverse mediators of Alzheimer's risk. *Nature* **603**, 885-892 (2022). <https://doi.org:10.1038/s41586-021-04369-3>
- 2 Allen, M. *et al.* Human whole genome genotype and transcriptome data for Alzheimer's and other neurodegenerative diseases. *Sci Data* **3**, 160089 (2016). <https://doi.org:10.1038/sdata.2016.89>
- 3 Bennett, D. A., Schneider, J. A., Arvanitakis, Z. & Wilson, R. S. Overview and findings from the religious orders study. *Curr Alzheimer Res* **9**, 628-645 (2012). <https://doi.org:10.2174/156720512801322573>
- 4 Wang, M. *et al.* The Mount Sinai cohort of large-scale genomic, transcriptomic and proteomic data in Alzheimer's disease. *Sci Data* **5**, 180185 (2018). <https://doi.org:10.1038/sdata.2018.185>
- 5 Korotkevich, G. *et al.* (2021). <https://doi.org:10.1101/060012>
- 6 Browaeys, R., Saelens, W. & Saeys, Y. NicheNet: modeling intercellular communication by linking ligands to target genes. *Nat Methods* **17**, 159-162 (2020). <https://doi.org:10.1038/s41592-019-0667-5>
- 7 Heneka, M. T. *et al.* Neuroinflammation in Alzheimer's disease. *Lancet Neurol* **14**, 388-405 (2015). [https://doi.org:10.1016/S1474-4422\(15\)70016-5](https://doi.org:10.1016/S1474-4422(15)70016-5)
- 8 Hur, J. Y. *et al.* The innate immunity protein IFITM3 modulates gamma-secretase in Alzheimer's disease. *Nature* **586**, 735-740 (2020). <https://doi.org:10.1038/s41586-020-2681-2>
- 9 Kano, A. *et al.* Endothelial cells require STAT3 for protection against endotoxin-induced inflammation. *The Journal of experimental medicine* **198**, 1517-1525 (2003). <https://doi.org:10.1084/jem.20030077>
- 10 Wu, W. *et al.* TLR ligand induced IL-6 counter-regulates the anti-viral CD8(+) T cell response during an acute retrovirus infection. *Sci Rep* **5**, 10501 (2015). <https://doi.org:10.1038/srep10501>
- 11 Ni, C. *et al.* Interferon-gamma safeguards blood-brain barrier during experimental autoimmune encephalomyelitis. *Am J Pathol* **184**, 3308-3320 (2014). <https://doi.org:10.1016/j.ajpath.2014.08.019>
- 12 Greenhill, C. J. *et al.* IL-6 trans-signaling modulates TLR4-dependent inflammatory responses via STAT3. *Journal of immunology (Baltimore, Md. : 1950)* **186**, 1199-1208 (2011). <https://doi.org:10.4049/jimmunol.1002971>
- 13 Deng, J. *et al.* IFN γ -responsiveness of endothelial cells leads to efficient angiostasis in tumours involving down-regulation of Dll4. *J Pathol* **233**, 170-182 (2014). <https://doi.org:10.1002/path.4340>
- 14 Middleton, K., Jones, J., Lwin, Z. & Coward, J. I. Interleukin-6: an angiogenic target in solid tumours. *Crit Rev Oncol Hematol* **89**, 129-139 (2014). <https://doi.org:10.1016/j.critrevonc.2013.08.004>
- 15 Nho, K. *et al.* Genome-wide transcriptome analysis identifies novel dysregulated genes implicated in Alzheimer's pathology. *Alzheimers Dement* **16**, 1213-1223 (2020). <https://doi.org:10.1002/alz.12092>
- 16 Skene, N. G. & Grant, S. G. Identification of Vulnerable Cell Types in Major Brain Disorders Using Single Cell Transcriptomes and Expression Weighted Cell Type Enrichment. *Front Neurosci* **10**, 16 (2016). <https://doi.org:10.3389/fnins.2016.00016>
- 17 Butovsky, O. *et al.* Identification of a unique TGF- β -dependent molecular and functional signature in microglia. *Nat Neurosci* **17**, 131-143 (2014). <https://doi.org:10.1038/nn.3599>
- 18 Zhang, Y. *et al.* Purification and characterization of progenitor and mature human astrocytes reveals transcriptional and functional differences with mouse. *Neuron* **89** (2016). <https://doi.org:10.1016/j.neuron.2015.11.013>

REVIEWERS' COMMENTS

Reviewer #1 (Remarks to the Author):

The authors have adequately addressed concerns I raised in the previous round of review. I have no further comments.

Reviewer #2 (Remarks to the Author):

The revised manuscript has seen significant improvement, yet some areas require further attention:

Figures 2 and 3 ought to be consolidated, given that they depict the DEGs and the associated GO terms. It would be beneficial to emphasize the top DE genes using violin or dot plots. Integrating Figure S13 or relevant figures at this juncture will provide a more holistic understanding of the transcriptomic alterations.

A predominant concern with this revision is the excessive relegation of data to the supplemental section. This data should be prominently positioned within the main figures. I encourage a review of figures from other articles affiliated with Nature; their information-dense presentations serve a purpose. It's imperative that substantial evidence isn't sidelined in supplemental sections, as this could risk oversight by the readership.

The figures representing the co-expression network analysis appear to lack depth. The objective should be to underscore both the modules and the co-expressed genes. I recommend the authors consider reconstructing these figures, drawing inspiration from Figure 2 in the following publication: <https://doi.org/10.1016/j.neuron.2020.11.002>.

Lastly, the NicheNet interactions would be best visualized using a circos plot, akin to the original NicheNet publication. This should also be incorporated into the main figure for greater clarity.

Reviewers' comments on the submission NCOMMS-21-41207A-Z with Authors' responses

Reviewers' comments with Authors' responses in **BOLD** and new text in the manuscript in *Italics*

Reviewer #1 (Remarks to the Author):

The authors have adequately addressed concerns I raised in the previous round of review. I have no further comments.

We would like to thank the reviewer for their comment.

Reviewer #2 (Remarks to the Author):

The revised manuscript has seen significant improvement, yet some areas require further attention:

Figures 2 and 3 ought to be consolidated, given that they depict the DEGs and the associated GO terms. It would be beneficial to emphasize the top DE genes using violin or dot plots. Integrating Figure S13 or relevant figures at this juncture will provide a more holistic understanding of the transcriptomic alterations. A predominant concern with this revision is the excessive relegation of data to the supplemental section. This data should be prominently positioned within the main figures. I encourage a review of figures from other articles affiliated with Nature; their information-dense presentations serve a purpose. It's imperative that substantial evidence isn't sidelined in supplemental sections, as this could risk oversight by the readership.

We would like to thank the reviewer for their positive evaluation and their precious suggestions. We have now extensively revised the figures describing the transcriptomic alterations of vascular cells to indeed provide a more holistic understanding of the transcriptomic alterations. We have thus consolidated Figures 2 and 3. The new Figure 2 describes the results of the differential gene expression analysis and the co-expression module analysis. Similarly, the new Figure 3 described the transcriptomic alterations in FB and PC. In accordance to the reviewer's suggestions, we have added a larger number of violin plots (eg Figure 2B, Figure 3B and 3D).

The new Figure 2 is shown below:

Figure 2 legend:

Figure 2 Alzheimer's disease is associated with dysregulation of vascular homeostasis in EC. (A) Volcano plot showing genes differentially expressed in AD relative to NDC donor cortical tissue in EC. Representative significantly differentially expressed genes are identified. (B) Violin plots of representative genes differentially expressed in EC with AD relative to NDC. *ANGPT2* (logFC=1.46, padj=0.04), *HIF1A* (logFC=0.68, padj=0.02), *MEF2C* (logFC=0.28, padj=0.09) and *FGF2* (logFC=0.34, padj=0.05) are significantly upregulated, whereas *RASAL2* (logFC=-0.76, padj=3.48x10⁻⁶), *IFNGR1* (logFC=0.89, padj=0.03), *ADAM10* (logFC=-0.30, padj=0.06) and *PICALM* (logFC=-0.27, padj=0.02) are downregulated. Statistical significance was determined using a likelihood ratio test with a mixed-effects model and a zero-inflated negative binomial distribution (two-sided). For demonstration purposes, the *FGF2* violin plot describes expression only for nuclei in which *FGF2* is expressed, although the statistical analysis was performed on all nuclei. (C) Dot plots of the functional enrichment analysis on the DEG that are up- and down-regulated in EC (dot size, functional enrichment gene set size; colour, FDR, one-sided overrepresentation Fisher's exact test) with AD relative NDC. (D) IHC of sections from the somatosensory cortex of NDC (left) and AD (right) donors highlighting increased expression of ANG2 (coded by *ANGPT2*), FGF2, FGFR1 and decreased expression of ADAM10 in the vessel wall with AD. Arrowheads denote the protein binding in the vascular wall. Scale bar=50 µm. The IHC experiment was performed on 24 independent samples. (E) Gene co-expression module hierarchy for EC. Modules that belong to the same branch are related, i.e., larger ("parent") modules are closer to the centre of the plot and are further divided into subset ("children") modules. "Children" modules are subsets of the "parent" ones and have higher numbers as names than their "parents". Modules that show a significant overrepresentation of DEG (as shown in the volcano plots of **Figure 2A**) by means of a (one-sided) Fisher's exact test are labelled and represented as coloured points in the graph (red, for modules showing an overrepresentation of upregulated DEG; blue, showing an overrepresentation of downregulated DEG). Module number font size corresponds to the significance of the overrepresentation of DEG in the module. In the boxes, the top (maximum 5) hub genes (genes with the higher number of significant correlations within the module) are

described. (F) Circular heatmap of odds ratios from the functional enrichment analyses for the EC modules that show a significant DEG overrepresentation (significant modules that show redundant functional enrichment terms were omitted from this heatmap). The adjusted p values of the significance of the overrepresentation are provided in Supplementary file 5. The inner two tracks of the circular heatmap represent the significance ($-\log_{10}(p_{adj})$, one-sided) of the overrepresentation of down- (innermost track) and up-regulated DEG (second innermost track). The DGE and co-expression analyses were performed on 70'537 nuclei from 77 independent samples. Source data are provided as a Source data file.

Figure 3:

Figure 3 legend:

Figure 3 Angiogenic and inflammatory pathways are differentially expressed in FB and PC co-expression network modules with AD. Volcano and violin plots showing genes differentially expressed in AD relative to NDC donor cortical tissue in FB (A-B) and PC (C-D). In FB, *PDE7A* (logFC=1.33, padj=0.007), *TRPM3* (logFC=1.44, padj=0.09), *ROBO1* (logFC=1.35, padj=0.09) are significantly upregulated, whereas *BCL2L1* (logFC=-0.35, padj=0.04), *SPTBN1* (logFC=-1.34, padj=0.0001) and *LAMC1* (logFC=-0.84, padj=0.01) are downregulated. In PC, *TCF4* (logFC=1.27, padj=0.01), *ARHGAP29* (logFC=1.47, padj=0.004), *PLOD2* (logFC=0.91, padj=0.003) are significantly upregulated, whereas *RASAL2* (logFC=-1.15, padj=0.005), *EGFR* (logFC=-0.76, padj=0.07) and *CFLAR* (logFC=-0.38, padj=0.03) are downregulated. Statistical significance was determined using a likelihood ratio test with a mixed-effects model and a zero-inflated negative binomial distribution (two-sided). For demonstration purposes, the *TRPM3*, *EGFR*, *PLOD2*, *LAMC1* and *BCL2L1* violin plots describe expression only for nuclei in which the respective genes are expressed, although the statistical analysis was performed on all nuclei. (E) Gene co-expression module hierarchy for FB and (F) PC. Modules that belong to the same branch are related, i.e., larger (“parent”) modules are closer to the centre of the plot and are further divided into subset (“children”) modules. “Children” modules are subsets of the “parent” ones and have higher numbers as names than their “parents”. Modules that show a significant overrepresentation of DEG (as shown in the volcano plots of **Figure 3A-C**) by means of a one-sided Fisher’s exact test are labelled and represented as coloured points in the graph (red, for modules showing an overrepresentation of upregulated DEG; blue, showing an overrepresentation of downregulated DEG). Module number font size corresponds to the significance of the overrepresentation of DEG in the module. In the boxes, the top (maximum 5) hub genes (genes with the higher number of significant correlations within the module) are described. (G) Circular heatmap of odds ratios from the functional enrichment analyses for the FB and (H) PC modules that show a significant DEG overrepresentation (significant modules that show redundant functional enrichment terms were omitted from this heatmap). The adjusted p values of the significance of the overrepresentation are provided in Supplementary file 5. The inner two tracks of the circular

heatmap represent the significance ($-\log_{10}(\text{padj})$, one-sided) of the overrepresentation of down- (innermost track) and up-regulated DEG (second innermost track). The DGE and co-expression analyses were performed on 9'594 PC and 20'885 FB nuclei from 57 independent samples. Source data are provided as a Source data file.

The figures representing the co-expression network analysis appear to lack depth. The objective should be to underscore both the modules and the co-expressed genes. I recommend the authors consider reconstructing these figures, drawing inspiration from Figure 2 in the following publication: <https://doi.org/10.1016/j.neuron.2020.11.002>.

Inspired by the reviewer's suggestion, we have now added circular heatmaps that provide information about the co-expression modules for EC (Figure 2F), FB (Figure 3G) and PC (Figure 3H). These circular heatmaps provide information about the functional enrichment of the prioritized modules. In addition, the overrepresentation of DEG in the prioritized modules is represented in the same heatmaps. Furthermore, we have included the co-expression network hierarchy plots to the main figures (2E for EC and 3E and 3F for FB and PC, respectively). These plots have now been enriched with the addition of the top hub genes of the most significant modules. In this way, according to the reviewer's suggestion, the reader obtains insight in the functional pathways that are overrepresented in the modules and also in the individual hub genes of the modules.

Lastly, the NicheNet interactions would be best visualized using a circos plot, akin to the original NicheNet publication. This should also be incorporated into the main figure for greater clarity.

We have followed the reviewer's advice and created a circos plot, combined with a circular heatmap. In this way we combine the information that the dotplots and the heatmaps of the previous version of our NicheNet figures presented in a more elegant and concise manner. We also include a heatmap of the characteristics of the target genes in the "reveiver" cell types in terms of their differential expression. The NicheNet analysis result for EC is included in the main figure 8 and the results for FB and PC in Figure S12.

Figure 8:

Legend for Figure 8:

Figure 8 NicheNet intercellular communication analysis identified potential regulators of EC DEG associated with proinflammatory and anti-angiogenic gene expression in astrocytes and perivascular macrophages. (A) Circular heatmap and chord diagram of the results of the NicheNet analysis. The circular plot is divided (based on the innermost track) to separately represented ligand (black track) and target genes (grey track). A heatmap for the ligand genes on the 2nd to the 6th tracks (from inner- to outer-most) represents the average scaled expression of each ligand in each of the “sender” vascular cell types (each cell type - $\log_{10}(\text{padj})$ is represented on a different track). The two outermost tracks describe the differential expression (\log_{FC}) of the associated genes in the “received” EC. The outermost track represented the \log_{FC} and the 2nd outermost track represents the value. The links of the diagram represents the regulatory potential between the ligand and the target genes. (B) Astrocytic *APOE* ($\log_{FC}=0.96$, $pval=.03$), PVM *VEGFA* ($\log_{FC}= -0.31$, $pval=0.08$), *TGFB1* ($\log_{FC}= -0.28$, $pval=0.0007$) and *GPNMB* ($\log_{FC}=0.23$, $pval=0.04$) potentially regulate the majority of EC DEG and are significantly differentially expressed with AD relative to NDC. (C)

Statistical significance was determined using a likelihood ratio test with a mixed-effects model and a zero-inflated negative binomial distribution. The analysis was performed on 170'299 astrocytic and 14'861 PVM nuclei from 57 independent samples. P values (unadjusted) refer to two-sided statistical tests. Source data are provided as a Source data file.

(C) IHC in a sample from the entorhinal cortex of sections from NDC (left) and AD (right) donors binding of VEGFA in the vessel wall with AD. Arrowheads denote the VEGFA binding in the vascular wall. The IHC experiment was performed on 24 independent samples. Scale bar= 50 μ m. Source data are provided as a Source data file.

Figure S12

Legend for figure S12

Figure S12 NicheNet intercellular communication analysis of potential regulators of FB

DEG (A) Circular heatmap and chord diagram of the results of the NicheNet analysis for FB

and (B) PC. The circular plot is divided (based on the innermost track) in ligand genes (black track) and target genes (grey track). The heatmap corresponding to the ligand genes from the 2nd to the 6th innermost tracks represents the average scaled expression of each ligand in each of the “sender” cell types of the vasculature (each cell type is represented on a different track). The two outermost tracks, which correspond to the target genes (grey track), represent the results of the DGE analysis of these genes in the “received” FB (A) or PC (B). The outermost track represented the logFC and the 2nd outermost track represents the $-\log_{10}(\text{padj})$ value. The links of the diagram represents the regulatory potential between the ligand and the target genes. Source data are provided as a Source data file.